# Improving Skin Cancer Treatment by Dual Drug Co-Encapsulation into Liposomal Systems—An Integrated Approach towards Anticancer Synergism and Targeted Delivery

**DOI:** 10.3390/pharmaceutics16091200

**Published:** 2024-09-12

**Authors:** Margarida Corte-Real, Francisco Veiga, Ana Cláudia Paiva-Santos, Patrícia C. Pires

**Affiliations:** 1Department of Pharmaceutical Technology, Faculty of Pharmacy, University of Coimbra, Azinhaga de Santa Comba, 3000-548 Coimbra, Portugalfveiga@ci.uc.pt (F.V.); 2REQUIMTE/LAQV, Group of Pharmaceutical Technology, Faculty of Pharmacy, University of Coimbra, 3000-548 Coimbra, Portugal; 3Health Sciences Research Centre (CICS-UBI), University of Beira Interior, Av. Infante D. Henrique, 6200-506 Covilhã, Portugal

**Keywords:** skin cancer, dual-loaded, liposomes, nanosystems, melanoma, squamous cell carcinoma

## Abstract

Skin cancer is a high-incidence complex disease, representing a significant challenge to public health, with conventional treatments often having limited efficacy and severe side effects. Nanocarrier-based systems provide a controlled, targeted, and efficacious methodology for the delivery of therapeutic molecules, leading to enhanced therapeutic efficacy, the protection of active molecules from degradation, and reduced adverse effects. These features are even more relevant in dual-loaded nanosystems, with the encapsulated drug molecules leading to synergistic antitumor effects. This review examines the potential of improving the treatment of skin cancer through dual-loaded liposomal systems. The performed analysis focused on the characterization of the developed liposomal formulations’ particle size, polydispersity index, zeta potential, encapsulation efficiency, drug release, and in vitro and/or in vivo therapeutic efficacy and safety. The combination of therapeutic agents such as doxorubicin, 5-fluorouracil, paclitaxel, cetuximab, celecoxib, curcumin, resveratrol, quercetin, bufalin, hispolon, ceramide, DNA, STAT3 siRNA, Bcl-xl siRNA, Aurora-A inhibitor XY-4, 1-Methyl-tryptophan, and cytosine–phosphate–guanosine anionic peptide led to increased and targeted anticancer effects, having relevant complementary effects as well, including antioxidant, anti-inflammatory, and immunomodulatory activities, all relevant in skin cancer pathophysiology. The substantial potential of co-loaded liposomal systems as highly promising for advancing skin cancer treatment is demonstrated.

## 1. Introduction

The complexity of tumorigenesis and the challenges associated with cancer treatment make cancer a serious threat to human health. According to the World Health Organization, it was estimated that, in 2022, there were approximately 20 million new cases of cancer and 9.7 million deaths that resulted from the disease. Furthermore, approximately 1 in 5 individuals developed cancer at some point during their lifetime. Of these cases, it was projected that 1 in 12 women and 1 in 9 men succumbed to the disease. The three most prevalent cancer types in 2022 were lung, breast, and colorectal cancers [1,2,3,4].

While there are well-characterized triggers in skin cancer, including prolonged sun exposure, indoor tanning, weak immunity, family history, and certain moles, numerous other factors remain undetermined [5,6]. The tumor microenvironment, which comprises a variety of non-cancerous cells, e.g., fibroblasts, endothelial cells, and immune cells, is known to influence cancer progression and treatment response [4]. It was estimated that over 1.5 million new cases of skin cancer occurred in 2022, with approximately 330,000 of these cases being diagnosed as melanoma worldwide. It was further estimated that nearly 60,000 individuals died from the disease. In the majority of world regions, the incidence of melanoma is higher in men than in women [7,8]. In 2022, New Zealand exhibited the highest overall mortality rate from skin cancer, while the United States of America and China demonstrated the highest incidence of deaths from skin cancer [9,10].

Currently, the treatment of skin cancer may be undertaken through the use of surgical procedures or medication. The treatment options for precancerous skin lesions or skin cancer depend on the size, depth, type, and location of the lesions. The standard procedures for the eradication of large-sized skin cancer in its initial stages include excisional and Mohs surgical procedures, as well as radiation therapy, with targeted therapy or immunotherapy. Nevertheless, the treatment of skin cancer (of small size) typically involves electrodesiccation and curettage, cryotherapy, and laser and light-based treatments, along with targeted therapy and immunotherapy. The objective of targeted therapy and immunotherapy in combination with other types of therapies is to prevent the recurrence of the tumor following its excision or elimination through physical techniques. Furthermore, in advanced metastatic phases of skin cancer, the administration of chemotherapeutic agents is strongly recommended. A concise overview of the currently applied skin cancer treatment procedures is diagrammatically represented in Figure 1, as well as their limitations. Although chemotherapy is an effective method for treating several types of cancer, chemotherapeutic agents do have some limitations. These include low bioavailability and solubility, as well as unfavorable pharmacokinetics with inadequate biodistribution, which can compromise their clinical application and lead to often severe adverse effects [5,6,11].

Regarding the several approaches employed to deliver therapeutic agents for skin cancer, nanomedicine has proven to be effective in overcoming the limitations of conventional therapies [5,6,11]. In order to overcome conventional skin cancer therapy limitations, researchers are increasingly investing in combination therapy. It is largely acknowledged that combination therapy can promote target selectivity and reduce cancer drug resistance, as well as inducing synergistic and enhanced therapeutic effects [12]. Nanotechnology has become a promising area of research in the field of skin cancer therapy, providing innovative solutions for targeted drug delivery and therapeutic interventions. In particular, nanocarriers offer a potentially powerful tool for enhancing the selectivity and efficacy of drug delivery to cancerous cells while reducing the incidence of dose-dependent side effects and avoiding the development of drug resistance [13,14,15]. Drug nanocarriers demonstrated several benefits, including the controlled release of active compounds, extended lifetime in systemic circulation, and minimization of normal cells’ toxicity, all while avoiding the reticuloendothelial system, as well as allowing multi-drug encapsulation [12,15,16,17]. Nanotechnology can also overcome the hydrophobicity issue of certain bioactive compounds by encapsulating them in a nanovesicle, improving both drug strength in formulation and their stability [12].

Additionally, it is well established that tumors can elicit immune responses, including the activation of self-reactive T-cells. These responses can be tolerated by the host, allowing the tumor to evade recognition and eradication by the immune system. In this context, and as mentioned, cancer therapy can also involve the immune system, the so-called immunotherapy, inducing and amplifying antigen-specific responses. This process may result in enduring immune memory, which can be employed to treat cancer. This advanced method includes tumor vaccines and immune checkpoint blockade. However, only a few patients benefit from this therapy, due to the minimal objective response rates. Immune checkpoints on the surfaces of cells regulate immune responses, inactivating T-cells and preventing them from damaging normal cells. However, cancer cells can be benefited by these checkpoints when they deactivate T-cells, allowing cancer cells to continue proliferating. Immune checkpoint inhibitors block these checkpoints, thereby allowing T-cells to attack cancer cells [18,19].

Overall, significant research and development efforts are currently underway to explore and improve nanotechnologies for the treatment of skin cancer. However, there are currently no commercially available dual-loaded nanosystems. In order to achieve the commercialization of the developed formulations, it is crucial to take special account of market requirements, specifically in regard to safety, efficacy, stability, and design, as well as sensory aspects in the case of topical administration and, of course, costs [20,21].

### 1.1. Types of Skin Cancer

About 90% of human malignancies are focused on epithelial cells [22]. Generally, skin cancer can be subdivided into two main categories: melanoma, which arises from a dysfunction in melanocytes, and non-melanoma skin cancers, such as basal cell carcinoma (BCC) and squamous cell carcinoma (SCC), derived from epidermal cells (Figure 2).

BCC is derived from basal epidermal keratinocytes, as well as eccrine sweat ducts and hair follicles. Additionally, BCC is dependent on surrounding stroma for sustenance and growth. As a result, the likelihood of metastasis through the blood or lymphatic systems is less than 1%. The primary risk factor for BCC is intermittent sun exposure, and the majority of BCC tumors occur in the head and neck, with a smaller percentage occurring on the nose [5,23,24]. On the other hand, SCC is the second skin cancer that is most frequently developed, following BCC. SCC is a markedly mutated human cancer, exhibiting greater aggressiveness, and has a propensity for more rapid proliferation and metastasis to various parts of the body in comparison to BCC. It is estimated that 95% of SCC cases are associated with mutations in the tumor suppressor gene TP53, which are predominantly induced by ultraviolet radiation (UVR). However, other factors play a role in the development of SCC, including human papillomavirus, genodermatoses, chronic lesions associated with inflammatory conditions, medications (e.g., tumor necrosis factor-α inhibitors), and non-healing wounds or scarring. SCC typically manifests on the head and neck and the dorsal surface of the hands and forearms [22,23,25].

Finally, melanoma cancer cells accrue from the modification of normal melanocytes. Even in an initial stage of melanoma development, the dysregulation of the cell cycle and uncontrolled cell proliferation might lead to metastases. Additionally, cutaneous melanoma can manifest de novo following intermittent, occasional, and intense UVR exposure. However, there is scientific evidence indicating that up to 30% of cases may be due to pre-existing regions of pigmentation derived from melanocytes, irrespective of whether these regions have been exposed to UVR. Melanoma cells demonstrate molecular modifications of the RAS/BRAF/MEK/ERK/MAPK signaling pathway, which mediates the aberrant proliferation of malignant melanocytes. Moreover, genetic variations in the Cyclin-dependent kinase inhibitor 2A (CDKN2A) gene, and melanocortin 1 receptor (MC1R) genetic polymorphisms are associated with an increased risk of developing melanoma. In early stages of melanoma, cure can reach a 90 to 100% rate of recovery, while cancer progression may decrease the probability of cure [26,27,28]. Despite the greater survival rate and prognosis demonstrated by the surgical removal of melanoma cells, in an advanced stage of metastasis, it does not prevent relapse. In fact, treatments such as radiotherapy, chemotherapy, and immunotherapy, alone or in combination, appear to be promising for metastatic melanoma treatment. Nonetheless, systemic therapies generally show high recurrence rates and have not been demonstrated to be effective strategies for melanoma treatment. In this regard, it is urgent to develop more efficient melanoma treatments towards widespread remission [15,26].

### 1.2. Cancer Signaling Pathways

A comprehensive understanding of the molecular and cellular pathways that underpin the development and progression of skin cancer is vital for the development of effective therapeutic strategies and preventive measures [29,30].

Survivin, the smallest member of the inhibitor-of-apoptosis protein (IAP) family, plays an important role in response to anticancer therapies by inhibiting apoptosis and promoting mitosis [31,32]. The upregulation of survivin expression contributes to the metastasis of several types of cancers, including skin cancer [33]. Besides its high expression in cancer, survivin is also expressed in some normal adult tissues, including the skin. It is mainly detected in the nucleus of keratinocyte stem cells (KSCs) but also in fibroblasts and melanocytes. The inhibition of apoptosis by binding caspases and blocking their activity is guaranteed by BIR (baculovirus IAP repeat), a protein domain used for the homodimerization of survivin and to interact with other chromosome passenger proteins. The coiled-coil α-helix domain of survivin allows the regulation of cell division [34,35]. The ability of survivin to regulate cell division is linked to its nuclear localization, whereas the mitochondrial localization of survivin is related with apoptosis inhibition. Nuclear surviving promotes cell exit from G1 checkpoint arrest and subsequent progression into the S phase. However, it does not prevent cell apoptosis. On the other hand, mitochondrial survivin is related with oncogenic transformation, potentially enhancing resistance to apoptosis in oncogenic cells by thwarting the activation of effector caspases [34,35].

Besides survivin, the epidermal growth factor receptor (EGFR) pathway is also of interest in cancer therapeutics. EGFR is a transmembrane receptor tyrosine kinase of the human epidermal growth factor receptor (HER) family, which is atypically activated in several epithelial tumors. The activation of EGFR can be assured by overexpression, ligand-dependent and ligand-independent mechanisms. EGFR ligands include transforming growth factor α and epidermal growth factor. The overexpression of EGFR through cancer cells leads to ligand-independent dimerization and activation. On the other hand, once linked, it induces a conformational modification in EGFR, promoting the activity of its tyrosine kinase. In a ligand-independent mechanism, the activation of EGFR can be assured by urokinase plasminogen. When EGFR is activated, multiple signaling pathways are triggered, and among them is the mitogen-activated protein kinase (MAPK) pathway, leading to cell proliferation, survival, and, occasionally, transformation. The phosphatidylinositol 3-kinase/protein kinase B (PI3K-AKT) pathway can be also activated by EGFR, regulating cell survival. This pathway is negatively controlled by the tumor suppressor phosphatase and tensin homologue (PTEN) gene, and its loss of function results in the increased concentration of phosphatidylinositol-3,4,5-triphosphate and successive AKT hyperactivation, protecting cancer cells from several apoptotic stimuli. In these pathways, the resulting transcriptional profiles are different from each other, triggering multiple cellular responses, including cell differentiation, proliferation, invasion, DNA repair, apoptosis, and survival (Figure 3) [29,36,37].

The interference of cell-cycle checkpoints has been widely recognized as a hallmark of cancer. Particularly, the dysregulation of G1-S transition mediated by the D-cyclin-dependent kinase (CDK) 4/6 pathway is observed in over 90% of melanoma cases. Thus, tumor cells predominantly depend on the G2-M checkpoint to stop the cell cycle for DNA damage repair. Once dysregulated, cancer cells start to proliferate at a fast rate [38]. The transcription of cell cycle proteins, including cyclin D and cyclin-dependent kinases (CDK), are controlled by the MAPK pathway, promoting cyclin D–CDK4/6 activity. Upon activation, cyclin D complexes with CDK4/6, which phosphorylates Retinoblastoma protein (Rb), and detaches the transcriptionally repressive Rb–E2F complex. E2F transcription factor induces cell-cycle progression, resulting in increased proliferation and survival [18,39].

Moreover, apoptosis is a form of regulated cell death that involves the activation of specific enzymes called caspases, which belong to the family of cysteine–aspartic enzymes. These proteases trigger the exposure of phosphatidylserine on the cell surface, the condensation of chromatin, the fragmentation of DNA, and the formation of apoptotic bodies, thereby controlling cell death. The internal pathway of apoptosis is characterized by the release of cytochrome c from the mitochondria and into the cytoplasm, leading to the activation of caspase-9. Caspase-9 then activates caspase-3 and caspase-7, which carry out the process of cell destruction [40]. Caspase-3 is known to be a key downstream effector or executioner protease involved in mammalian cell apoptosis. The development of the apoptotic phenotype is completed with the activation of caspases [40,41].

### 1.3. Challenges Associated with Administration Routes

The administration route of the therapeutic formulation can influence its biodistribution in tumors. In this regard, the skin serves as a barrier for two distinct forms of drug delivery: topical and transdermal. All topical and transdermal drug formulations are applied onto the skin, and are therefore highly advantageous in the avoidance of contact with the gastric fluids, intestinal fluids, systemic circulation enzymes, and hepatic first-pass drug metabolism. Topical formulations are meant to only slightly penetrate the skin, which is their intended mode of action, creating a localized effect. However, only transdermal systems are designed to cross the skin barrier and exert their therapeutic effects on deep or distant tissues, potentially reaching the systemic circulation. Hence, transdermal administration is useful for both systemic and local effects. In order to achieve a successful transdermal administration, it is of the utmost importance to consider the physicochemical properties of the drug in question, including its lipid solubility and molecular weight. Here, microneedles can be used, a non-invasive physical enhancer of skin penetration, which in addition to a negligible risk of infection, do not necessitate medical expertise for administration [20,42,43].

It is noteworthy that healthy *stratum corneum* hampers the diffusion of topical formulations through the skin, contrary to damaged skin, which describes pre-cancerous/cancerous skin conditions. Hence, in these and other pathological cases, skin permeation might be enhanced. Furthermore, the connective tissue’s extracellular matrix is supported by certain elements produced by fibroblasts, including collagen, glycosaminoglycans, and glycoproteins. In the absence of these cells, epidermis cells would not be able to proliferate through wound sites, preventing their regeneration. In this regard, fibroblasts have an important role in wound healing, which is a process often associated with skin cancer lesions [16,44].

It is also important that the drug delivery system prevents the drug’s systemic circulation elimination and enhances its localized accumulation in tumor cells as much as possible. Topical administration may not be completely satisfactory in this context due to the presence of keratinized cells localized on the tumor surface, preventing the penetration of the delivery system through deeper layers of the skin tumor. Here, iontophoresis can be adopted to overcome this obstacle, leading to a higher and deeper penetration of topically administered drugs by electro-osmosis and electromigration methods [36,37].

On the other hand, subcutaneous delivery is conducted in the hypodermis. Despite the proximity of the injection site to blood vasculature, not all therapeutic agents are effectively and uniformly delivered, resulting in a minimal effect on the elimination rate when compared to intravenous administration. Specific adverse effects associated with subcutaneous delivery include injection-site reactions, such as pain and erythema [45,46].

Moreover, when talking about systemic administration, it is important to note that the administration of an isolated drug for cancer treatment has led to several severe side effects, such as kidney failure, neuropathy, cardiac toxicity, neutropenia, alopecia, and myelosuppression. These side effects can be exacerbated due to the incapacity of localized drug accumulation at the tumor site, consequently requiring a higher dose of drug to reach a therapeutic level. Besides these adverse effects, single systemic drug administration therapy has proven to have other relevant limitations, including poor bioavailability, fast renal clearance, and multidrug resistance. In this regard, as mentioned, the combination of chemotherapeutic drugs can be proposed, based on different mechanisms of action, reaching an enhanced efficacy and synergistic therapeutic effects. Furthermore, nanometric delivery systems have shown to be revolutionary, allowing a greater cell penetration and tumor targeting [15,36].

### 1.4. Nanosystems for Skin Application—A Focus on Liposomes and Derived Systems 

It is widely acknowledged that nanosystems between 5 and 200 nm in diameter are suitable for tumor targeting. However, this assertion is contingent upon the specific characteristics of the tumor in question. The morphology, surface charge, and composition of nanoparticles have been described as highly variable and dependent on the properties attributed to them (e.g., stability, cell penetration, and toxicity). Nanosystems may be used for a sustained and regulated delivery of both hydrophobic and hydrophilic tumor-targeting molecules. The passive targeting of tumors may be achieved through the enhanced permeability and retention (EPR) effect, while active targeting may also be facilitated by nanosystems. The use of nanostructured materials enables precision-targeted drug delivery, which is a promising avenue of research in the treatment of melanoma. Furthermore, these nanosystems have the capacity to deliver therapeutic agents directly into malignant cells, sparing healthy cells and thus enhancing therapeutic efficacy while reducing adverse effects. Moreover, the use of nanosystems has the potential to both reduce the total dose needed for anticancer therapy and minimize the toxic off-target adverse effects associated with conventional treatments. The encapsulation, adsorption, or covalent coupling of anticancer biomolecules to nanocarriers can facilitate overcoming biological and physicochemical barriers. Additionally, the use of nanosystems allows for a synergistic integration of diverse anticancer therapeutic agents [27,28,47,48,49].

However, there is no therapy without limitations, and the potential limitations of nanocarriers in targeted drug delivery include drug resistance in cancer cells, which may influence the efficacy in cancer treatment; toxicity, dependent on composition, shape, size, and surface characteristics; the challenge of large-scale manufacturing and ensuring safety; high cost; regulatory issues; ethical concerns; and a lack of robust knowledge. All these issues must be taken into account when developing a nanosystem for therapeutical applications, and intense research is ongoing to address them [50,51]. 

The most extensively researched nanosystems with potential for use in skin cancer therapy include inorganic nanoparticles (metal nanoparticles, carbon nanotubes, and nanofibers), polymer-based nanoparticles (polymeric nanoparticles and dendrimers), and lipid-based systems (solid lipid nanoparticles, liposomes, ethosomes, and niosomes) [49,51]. In the context of lipid-based nanosystems, lipid nanocarriers are a particularly prospective area of research due to their distinctive structure and composition. These nanometric structures can solubilize and deliver therapeutic agents efficiently, thereby improving their therapeutic potential and bioavailability. 

The most commonly developed lipidic nanosystem type is liposomes, which are formed from amphoteric lipid membranes with a hydrophobic outer shell and a hydrophilic core, which enables them to incorporate both hydrophobic and hydrophilic compounds. Due to their composition, which is similar to that of cell membranes, liposomes are quite non-toxic, non-immunogenic, and overall biocompatible nanocarriers. The encapsulation of drugs within liposomes provides protection from degradation, thus avoiding premature exposure to the surrounding environment. In this way, the encapsulation of drugs within liposomes ensures the avoidance of drug accumulation in non-target organs [49]. Furthermore, the enhanced EPR mechanism of liposomes renders them a safe material [15]. Many investigations have confirmed that liposomes can enhance chemotherapeutic agents’ solubility and facilitate the retention of their bioactivity at the tumor site [19]. It is noteworthy that liposomes in the circulation system are able to act in circulation cancer cells, preventing them from inducing metastasis in other tissues [26]. Additionally, the usefulness of liposomes and derived nanosystems for cancer therapy has been further supported by recent patents. Lambros et al. [52] developed liposomes containing a cystine molecule as the targeting component, with the aim of achieving the targeted intracellular delivery of a therapeutic antitumor agent, such as peptides, proteins, antibodies, nucleic acids, siRNA, or other relevant organic or inorganic molecules, showing representative in vitro results, for human lung cancer (A549 cell line), and in vivo results, in a pancreatic tumor model (Pan02 cell line). In a different patent, Hong et al. [53] proved that liposomes loaded with irinotecan and having 1,2-distearoyl-sn-glycero-3-phosphorylethanolamine (DSPE), 1,2-distearoyl-sn-glycero-3-phosphocholine (DSPC), PEG 2000, and/or cholesterol in their composition were effective against an in vivo breast cancer model (human BT-474 cell xenografts) and colon cancer model (human HT-29 cell xenografts) through intravenous administration. Specifically for skin application, another patent [54] developed a paclitaxel-loaded ethosomal gel, generally containing cholesterol, a low-molecular-weight alcohol, a phospholipid, a stabilizing agent, and an antioxidant agent. Nevertheless, it appears to be hard to find recent patents on the use of these nanocarriers solely for skin cancer treatment, leaving room for further research [55].

There are several methods of liposomes and liposome-derived vesicle preparation, including the most common thin-film hydration (TFH), reverse-phase evaporation and ethanol injection [43]. TFH represents the most commonly employed methodology for the preparation of multilamellar vesicles (Figure 4). The preparation of these nanocarriers typically involves the dissolution of a lipid mixture (phospholipids, cholesterol, or other lipids) that might contain a hydrophobic drug molecule in chloroform or another organic solvent. Subsequently, the solvents are evaporated, forming a liposomal thin film. The thin film is then immersed in water, an aqueous solution, or a buffer that might contain a hydrophilic drug molecule to be included in the nanovesicles, resulting in a solution of multilamellar nanosystems with a usually large range of sizes. Extrusion and sonication can be employed to homogenize these lipid-based systems’ sizes [56,57]. In the reverse-phase evaporation method, the lipids, which have been dissolved in an appropriate solvent, are incorporated into an aqueous solution. Subsequently, the organic solvent volatilizes, resulting in the complete removal of the organic solvent. This process yields an aqueous liposomal suspension [56,58]. Finally, in the ethanol injection method, an ethanol-based lipid solution is introduced into a water-based solution, and the hydrophilic fraction is heated. Once the carriers have been obtained, the ethanol evaporates, leading to narrow-size-distribution nanocarriers [56,59].

Liposomes are regarded as an effective drug delivery system due to their capacity to accommodate a diverse range of drug characteristics. There are two types of drug loading in liposomes, the passive and active methods, where the therapeutic agent is encapsulated within the liposome during and subsequently to the preparation stage, respectively. The dispersion of hydrophilic drugs occurs in the aqueous phase, within the core of the liposomes. In contrast, hydrophobic agents are confined to the interior of the liposomal bilayer. On the other hand, active loading, also known as remote loading, involves the generation of a transmembrane ion or pH gradient that effectively propels the therapeutically active agent through the lipid membrane, resulting in loading efficiencies of up to 100% in certain cases [60,61].

In terms of structure, liposomes are usually spherical, monolayered, or multilayered vesicles that are formed through the self-assembly of diacyl-chain phospholipids (lipid bilayer) in aqueous solutions. The lipids employed in the preparation of liposomes can be classified into two main categories: natural lipids, derived from sources such as soya bean or egg yolk; synthetic lipids, including DPPC, DSPE, and 1,2-distearoyl-sn-glycero-3-phosphocholine (DSPC); synthetic phospholipids made from 1,2-dioleoyil-3-trimethylammonium propane (DOTAP) or 1,2-dioleoyil-sn-glycero-3-phosphoethanolamine (DOPE); steroid molecules, including cholesterol; and/or surfactants, such as sodium cholate (SC), amongst others [62]. A schematic representation and summary of liposomes’ most common preparation techniques, general structure, and physicochemical, therapeutic, and safety studies is present in Figure 5.

In what concerns how liposomes might exert their therapeutic action, this is directly related to how and where they release their cargo. Although drug release from liposomal systems can be non-specific, recent efforts have been made to control this release as much as possible, either by making the liposomal systems controllable by external stimuli, namely, by using compositions that are responsive to electric, light, thermal, magnetic, or other energy-sourced stimuli, or by making them answer to the tumor’s microenvironment, namely, changes in physiological signals such as pH, oxygen levels, ATP levels, or redox potential, and even the presence of specific enzymes [63,64]. Once the therapeutic active sites are reached, liposomes can release the drug either outside the cells, with the exogenous or endogenous trigger stimuli either disrupting the liposomal membrane or increasing its permeability to allow for the encapsulated drug molecule to be released, or after entering them, which is especially relevant when delivering genetic material, such as DNA or RNA, and which depends on vesicle composition, especially surface functionalization [62,63].

As for liposomal delivery methods, inherently, as mentioned, their composition allows them passive permeation through the biological membranes due to being formed by amphoteric lipids similar to those present in the biological membranes, as well as many times having in their composition permeation enhancers, such as ethanol (ethosomes) or surfactants (niosomes). Nevertheless, some methods might be used to increase penetration through the different biological barriers by using pretreatment or combination with ultrasound, iontophoresis, electroporation, or microneedles in the case of topical or transdermal administration [37,65,66,67].

### 1.5. Relevant Nanometric Formulations’ Physicochemical Characterization Parameters

The stability of nanosystems is crucial for ensuring their effective and safe application, depending on the techniques of formulation manufacturing. The most common and relevant nanosystem physicochemical characterization parameters are average particle size, polydispersity index (PDI), zeta potential (ZP), and encapsulation efficiency (EE), as well as in vitro drug release.

The size of the nanovesicles is a crucial factor that determines the in vivo fate of these structures. Therefore, it is essential to monitor this parameter with great care [43]. As a critical attribute of nanosystems, particle size affects cellular uptake, stability, encapsulation efficiency, and drug release profiles. If the particle size is small, the surface-to-volume ratio will be higher than with a larger vesicle. The vesicle size of lipidic carriers has been shown to have a meaningful influence on the delivery of bioactive compounds into the skin. Generally, a diameter higher than 600 nm blocks the delivery of the encapsulated drugs into deeper layers of the skin [68]. Based on the size of the vesicles, liposomes can be classified into small unilamellar vesicles (SUVw), large unilamellar vesicles (LUVs), and giant unilamellar vesicles (GUV), having sizes of 30–100 nm, 100–1000 nm, and 1–100 μm, respectively [56,62].

PDI is used to define the size range of the nanosystems. The PDI value can be between 0.0 and 1.0, from homogeneous sample to a highly polydisperse sample, respectively. Lipid-based carriers used in drug delivery, such as liposome formulations, are usually considered to be acceptable when having a PDI ≤ 0.3, which indicates a homogenous system of nanovesicles [68,69].

The stability of a nanoformulation can be predicted by the magnitude of the ZP, although it is not an absolute criteria of nanoparticle stability. This parameter has an important role in stability, protein interactions, circulation time, biocompatibility, and the permeability of the nanosystems. ZP is affected by temperature, pH, and ionic strength. At high absolute values of ZP, particles are highly charged, which prevents aggregation through electrostatic repulsion. On the other hand, low values indicate a higher risk of particle aggregation. Liposomes should have a |ZP| > 30 mV to avoid aggregation. Lower values of ZP can induce unstable systems with larger complexes due to the fast aggregation. Additionally, the size, shape, and charge of a nanosystem influence its cellular uptake due to the cell and tissue binding processes affected by ZP. In general, the higher the ZP, the stronger the membrane bindings and the greater the cellular uptake [70,71].

The percentage of drug retained inside a liposome, or derived nanosystems, known as encapsulation efficiency (EE), can be determined directly and indirectly. The direct method is based on the quantification of the payload of the nanoparticles (Equation (1). The indirect method allows for the quantification of the non-encapsulated drug (Equation (2) [59,72].
(1)$$\mathrm{Encapsulation}\;\mathrm{efficiency}\;(\%)=\frac{Concentration\;of\;encapsulated\;drug}{Concentration\;of\;total\;drug}\times 100$$
(2)$$\mathrm{Encapsulation}\;\mathrm{efficiency}\;(\%)=1-\frac{Concentration\;of\;free\;drug}{Concentration\;of\;total\;drug}\times 100$$

On the other hand, the release profile of a particular drug in the surrounding environment over a specific period of time is crucial to predict the bioavailability and, subsequently, the efficacy of the treatment. In the context of skin cancer, it is of paramount importance that a pharmaceutical agent is released in an efficacious manner and that the requisite quantity reaches the tumor site in order to exert its biological effects. The optimization of drug release and bioavailability facilitates the enhancement of therapeutic efficacy, the minimization of adverse effects, and the improvement of treatment adherence. Among the various techniques employed to evaluate drug release from nanostructures, the dialysis method is regarded as the most widely used approach. Regular dialysis and reverse dialysis, as well as side-by-side dialysis set-ups, can be employed. In the conventional dialysis technique, the therapeutic agent released from the nanocarriers diffuses across the dialysis membrane into the external compartment, where it is then sampled for quantification. Alternatively, in reverse dialysis, the nanosystems are located in the external compartment, while the internal section is sampled for therapeutic agent release. In the third-mentioned technique, side-by-side dialysis, both the receiver and donor cells are separated by a dialysis membrane, and sampling occurs from vertical Franz diffusion cell and the receiver cell. Dialysis cells are predominantly employed to investigate the release of small molecules from nanocarriers, rather than microcarriers, given that more straightforward techniques can be utilized for larger carriers. On the other hand, Franz cells and Ussing chambers have the advantage of enabling the evaluation of drug transportation across a range of epithelial tissues, including those derived from both human and animal biopsy samples, but, in this case, drug permeation assays, rather than drug release assays, are performed [73,74,75,76].

## 2. Dual-Loaded Liposomal Systems for the Treatment of Skin Cancer

The objective of this review is to provide a comprehensive and critical evaluation and summarization of the current research on the enhancement of skin cancer treatment through the drug co-encapsulation into liposomes and liposome-derived nanosystems. The assessment places particular emphasis on the effectiveness of combining two therapeutic agents within a single nanocarrier system. In addition to evaluating the stability, efficacy, and safety of this approach, the review also assesses the potential for reduced adverse effects and the possibility of synergistic action between the therapeutic agents when compared to isolated agents.

Firstly, the review examines the selection criteria of the therapeutic agents employed in the treatment of skin cancer, as well as the preparation methods used for the co-encapsulation of active agents. Moreover, the physicochemical parameters of the delivery systems are compared between the different experimental studies, thus enabling the obtaining of deeper insights into their potential for sustained and targeted drug delivery. Furthermore, the review investigates the cellular uptake, skin penetration, and in vitro and in vivo efficacy of dual-loaded nanocarriers, evaluating how these systems enhance therapeutic effects, including augmented cancer cell cytotoxicity, with the induction of apoptosis and other mechanisms, and overall antitumor activity in preclinical models.

Thus, this review will assess the preclinical relevance and potential for translating these advanced drug delivery systems into practical therapeutic solutions. The overall objective is to provide a comprehensive understanding of the current advancements in dual-loaded liposome-based systems and their impact on skin cancer treatment. The findings are meant to contribute to identifying the most promising strategies for enhancing the efficacy and safety of skin cancer therapies through innovative drug delivery systems. The following subsections will describe the several studies carried out in the context of the development of liposomes and liposome-derived systems with two simultaneously encapsulated therapeutic agents (summary in Figure 6). 

### 2.1. Doxorubicin and Celecoxib

Ahmed et al. [43] investigated doxorubicin and celecoxib co-loaded liposome administration, with pre-treated Derma roller^®^ microneedles, on melanoma skin. Due to the poor skin penetration of doxorubicin, investigators have been studying delivery systems to enhance drug skin penetration and its accumulation in the tumor site, as well as to prevent its systemic biodistribution. On the other hand, celecoxib is a non-steroidal anti-inflammatory drug (NSAID), and, in addition to enhancing the efficiency of other therapeutic agents, it also prevents skin cancer development, thus having intrinsic anticancer activity as well. Thus, the combination of these two therapeutic molecules in the same drug delivery system was meant to provide higher anticancer efficacy through synergistic effects. 

The liposomes were prepared by TFH, ethanol injection, reverse-phase evaporation, combined with the pH gradient method, and remote loading. The lipid phase was composed of hydrogenated soya bean phosphatidylcholine, where celecoxib was added. The application of ultrasonication resulted in a reduction in particle size. Subsequently, doxorubicin was added. A drug-loaded liposome gel was prepared, using the gelling polymer Carbopol. The objective of utilizing this component was to create an efficacious dermal base for topical administration. Particle size, PDI, and ZP were evaluated by cumulative analysis. EE was detected directly by UV spectroscopy, particle morphology was detected by TEM (Figure 7B), and in vitro drug release was performed using dialysis. The results revealed a small particle size (Figure 7A), and the PDI value indicated a homogeneous distribution (<0.3). The ZP value indicated that the liposomal suspensions were electrostatically stabilized. Thus, the authors successfully produced nanovesicles of a uniform shape and size. Additionally, all liposomal gel formulations exhibited a sustained and comparable in vitro release of doxorubicin and celecoxib, including co-loaded and free drug liposomes. Skin penetration studies were also performed in abdominal skin from female BALB/c nude mice. Microneedle pre-treatment followed by the administration of the co-loaded liposomal gel demonstrated higher drug penetration through the skin when compared to passive delivery. Cytotoxicity was evaluated on B16 cells (murine melanoma) by the MTT assay. The cytotoxicity effect was greater in the co-loaded nanometric formulation when compared to the free drug. In what concerns rheological properties, the co-loaded liposomal gel demonstrated high elasticity with a pseudoplastic behavior. 

Furthermore, the in vivo antitumor effect (Figure 7C–F), evaluated in a mouse xenograft model, was significantly higher after the administration of the developed co-loaded liposomes than with single drug liposomes. The administration of microneedles prior to the application of the gel also resulted in a notable improvement in tumor inhibition, being characterized by the formation of micro-holes in the *stratum corneum*, which facilitated drug permeation. Tumor weights of animals pre-treated with the microneedles showed smaller size compared to the untreated group. The mean weight of the mice did not exhibit a discernible change, and there were no animal deaths during the course of the treatment, indicating that co-loaded liposomes delivered by microneedles are safe to be administered in vivo. The authors suggested that the designed formulation is a promising method for the treatment of skin cancer, with efficiency on targeting inhibition, as well as insignificant adverse effects.

### 2.2. Doxorubicin and Ceramide

Chen et al. [17] developed co-loaded liposomes containing doxorubicin and ceramide for melanoma treatment to achieve an enhanced cytotoxic effect. The nanovesicles were prepared with three types of ceramides, including C6-ceramide, C8-ceramide, and C8-glucosylceramide. Ceramides have a signaling and structural role within the cell and in its membranes, respectively. These specific types of ceramides are distinguished by the carbon lengths, as well as the glucosyl change. As biological active compounds, ceramides can interfere in cell differentiation, cell cycle blocking, and cellular apoptosis, suggesting their potential as chemotherapeutic agents. In cancer, these ceramides have been proven to act through the PI3K/AKT pathway, by AKT dephosphorylation, increasing cytotoxicity and cell apoptosis, showing efficacy when co-loaded with other chemotherapeutic agents as well. However, the hydrophobicity of ceramide limits its clinical use, which can be overcome by liposomal encapsulation. On the other hand, doxorubicin, a well-known chemotherapeutic agent, as a free drug diffuses slowly through cell membranes, therefore having limited cancer cell uptake. Its encapsulation into liposomes might solve these issues. Additionally, the authors designed a formulation to reduce the necessary dose of doxorubicin and its dose-related adverse effects.

In this study, liposome formulations were prepared by TFH, followed by extrusion through a 100 nm polycarbonate filter, and subsequent remote loading with ammonium sulphate. Lipids such as DOTAP, DSPE, DPPC, and DSPC were used. The low and variable oil/water partition coefficient of amphiphilic doxorubicin demands a drug loading method that is gradient-dependent to better encapsulate doxorubicin in the aqueous core of the liposomes. In this regard, and to achieve an efficient and stable liposomal formulation encapsulation, the authors performed an active loading method based on the gradient of transmembrane ammonium sulphate. This was proven by the fact that doxorubicin was not encapsulated in liposomes in the absence of a pH gradient. As for the ceramide, it was integrated in the lipid layer, while doxorubicin was encapsulated in the aqueous compartment of the liposome. The final formulation was developed with doxorubicin and ceramide with an 8% molar ratio each, at a 5:1 M ratio of lipid drug.

The authors evaluated the particle size, size distribution, ZP, doxorubicin EE, and doxorubicin drug release. EE was determined through the direct method, particle size was measured by DLS, and ZP was determined by electrophoretic light scattering. It was proven that ammonium sulphate enhanced the EE efficiency, and, from a certain point, higher concentrations of doxorubicin resulted in a stagnation of the EE due to the saturation of the encapsulation procedure. Due to the potential of the rigidity improvement of the lipid layer induced by the presence of cholesterol, and the consequent enhanced physical stability of the resulting liposomes, the cholesterol impact on the liposomes’ characteristics was also evaluated. The results showed that that an increased percentage of cholesterol enhanced the EE. Thus, cholesterol had an important role in drug-loaded liposomes preparation by contributing to a better drug loading. Particle size was similar between the formulations in the presence or absence of ceramides, revealing that particle size and doxorubicin EE were not affected by the addition of these compounds. Therefore, the authors suggested that ceramides with different structures did not condition the liposomes’ drug encapsulation. Additionally, PDI values lower than 0.2 revealed a narrow size distribution for the co-loaded liposomal formulations.

The profile of doxorubicin in vitro drug release from the produced nanovesicles was evaluated through the dialysis technique. When compared to the free drug, the lipid bilayer of liposome formulations induced a significantly slower and more controlled release profile, which was deemed as desirable. Different ceramides in the formulations demonstrated similar release profiles. In fact, all the lipidic elements of the liposomal formulation contributed to a slower release profile of the drug. Additionally, the MTT assay allowed to evaluate the cytotoxicity effect of the optimized liposomal formulations on B16BL6 cells (murine melanoma). DOTAP/C8-ceramide liposomes, DOTAP/C8-glucosylceramide liposomes, DOTAP/C6-ceramide liposomes, and a drug solution (control) revealed the ascending order of cell viability. Additionally, ceramide and doxorubicin co-loaded liposomes demonstrated a much higher fluorescence when compared to no ceramide formulations or doxorubicin solutions. Thus, in summary, liposomes with C8-ceramide and doxorubicin, with DOTAP, showed higher cytotoxicity and cellular uptake in B16BL6, especially when compared to doxorubicin carriers without ceramide. This was explained by the capacity of ceramide to adapt the lipid bilayer, thereby improving the permeation and subsequent uptake of doxorubicin into cancer cells. Thus, this study led to the development of small, homogeneous co-loaded liposomes, with controlled drug release and increased therapeutic efficacy in an in vitro melanoma model, proving to be an advantageous alternative to current conventional treatments, with the synergy being evident between a known chemotherapeutical drug, doxorubicin, and ceramides.

### 2.3. Doxorubicin and Hispolon

Al Saqr et al. [15] also studied the effect of co-encapsulated liposomes, with hispolon and doxorubicin, in melanoma treatment. Derived from the fungi *Phellinus linteus*, the polyphenolic compound hispolon has proven anti-inflammatory, antioxidant, anti-proliferative, and anti-metastatic properties. In cancer, hispolon is known for inducing apoptosis through reactive oxygen species (ROS) induction and for blocking cell cycle through PI3K/AKT and ERK pathways. It also inhibits B cell lymphoma-2 protein (Bcl-2) and complexes I and IV and promotes the expression of caspase enzymes and the bcl-2-associated protein x (Bax) gene, as well as lipid peroxidation and nitrite content levels associated with apoptosis. Additionally, as mentioned, the anthracycline doxorubicin is a known chemotherapeutic drug, and its anticancer properties stem from its ability to inhibit topoisomerase II, intercalate between DNA base pairs in the double helix, and subsequently disrupt the replication and transcription of DNA. Despite the relevance of doxorubicin as a chemotherapeutic molecule, this drug has a narrow therapeutic window, and, in severe toxicity situations, doxorubicin can lead to bone marrow toxicity and cardiotoxicity because of its inability to selectively target abnormal cells only. The combination of chemosensitizers with doxorubicin has been proposed to inhibit multidrug resistance, minimize myocardial impairment, and increase apoptosis. Compared to the isolated drug, dual-drug therapy has been shown to induce enhanced overall and total remission rates. Hispolon, as a lipophilic compound, can be localized within the liposomal lipid bilayer, and doxorubicin, as a hydrophilic compound, can be encapsulated in the aqueous core. Again, the synergic activity of both bioactive compounds was an important focus.

Regarding the preparation method of developed liposomes, firstly, the authors performed the TFH method. However, the results were not as expected or intended, since the produced vesicles led to low EE values for both drugs. To increase the EE, hispolon-loaded liposomes were produced by using the remote film loading method, and an active loading technique was conducted for doxorubicin-loaded liposome preparation. Additionally, since the solubility of doxorubicin is influenced by pH because of its amino group, it was encapsulated in an acidic environment, where it is more soluble. Formulation composition included DSPC, cholesterol, and 1,2-distearoyl-sn-glycero-3-phosphoethanolamine-N-[methoxy (polyethylene glycol)-2000] (ammonium salt), which were dissolved in chloroform. Lipid-thin films were hydrated with ammonium sulfate (pH 5.5). Unencapsulated ammonium was removed by dialysis against a sucrose solution. The bioactive compounds were then encapsulated. 

The characterization of the formulations was based on particle size, PDI, EE, and the drug release profile of isolated drug liposomes. Particle size was analyzed by DLS, ZP was determined by photon correlation spectroscopy, osmolality was analyzed by vapor pressure osmometry, and EE was determined by spectrophotometry (direct method). The in vitro drug release profile was determined by dialysis, and the quantification of each drug was determined by spectrophotometry. According to the results, the liposomes revealed a small particle size, monodisperse distribution (low PDI values) and potentially stable vesicles due to high absolute ZP values. High drug EE and a suitable osmolality (~280 mOsm/Kg) proved that the formulations were appropriate for intracellular drug delivery. In what concerns drug release assays, due to the liposomal bilayer barrier, the drugs required more time to be released when compared to the control (drug solutions), demonstrating a slower, controlled release profile for both active compounds. The slower release of doxorubicin compared to hispolon may be justified by a pH gradient-dependent crystallization.

In vitro cytotoxicity studies were also conducted, in B16BL6 cells (melanoma), to analyze not only overall cytotoxicity but also cellular uptake and apoptosis. Cytotoxicity was measured using an MTT assay. Co-loaded liposomes revealed higher cytotoxicity when compared to free drug solutions and increased apoptosis when compared to monotherapy. The authors verified the absence cytotoxicity for blank liposomes, which suggested that the lipids were well tolerated by the B16BL6 cell line. 

Hence, in this study, the chemotherapeutic agents demonstrated a synergistic action by inhibiting B16FL6 cells. The produced nanometric vesicles should be able to promote drug targeting, prevent uptake by the reticular endothelial system (due to the protective charged layer surrounding the liposome), leading to a prolonged retention in the circulation, and lead to an overall improved therapeutic efficacy.

### 2.4. 5-Fluorouracil and Cetuximab

Petrilli et al. [37] investigated the efficacy of EGFR-targeted liposomes loaded with the chemotherapeutic agent 5-fluorouracil and the antibody cetuximab for the treatment of SCC (Figure 8A). Cetuximab, an IgG1 monoclonal antibody, can bind to EGFR, leading to its inhibition and resulting in cell cycle arrest and reduced angiogenesis, cell proliferation, and metastasis. In addition to its cytotoxic effect, cetuximab can also increase the activation of pro-apoptotic molecules through synergistic effects with other chemotherapeutic agents and/or radiotherapy. As for 5-fluorouracil, it is a well-known and reasonably effective chemotherapeutic agent, but it often leads to severe side effects due to its lack of targeting. The aim of cetuximab co-administration was to improve cancer cell targeting and overall treatment efficacy due to its highly specific affinity of cetuximab for EGFR, an overexpressed receptor in SCC cells, and the consequent internalization and degradation of EGFR.

Prior to conjugation with the liposomes, cetuximab was thiolated by the addition of Traut’s reagent dissolved in a PBS/EDTA buffer. Then, the authors used TFH to prepare the liposomes, using the lipids DSPC and cholesterol, as well as PBS pH 7.4, containing 5-fluorouracil. Cetuximab-loaded immunoliposomes were prepared by the same method but with the addition of 1,2-distearoyl-sn-glycero-3-phosphoethanolamine-N [maleimide (polyethylene glycol)-2000] (ammonium salt). Thiolated cetuximab allowed cetuximab to be attached to the maleimide moiety on the surface of the liposome.

The characterization of the developed liposomes and immunoliposomes included the analysis of particle size, PDI, ZP, and 5-fluorouracil EE. Immunoliposomes showed no significant differences in particle size, PDI, and 5-fluorouracil EE compared to regular liposomes. However, immunoliposomes showed a higher ZP modulus than regular liposomes, which meant that they had the propensity for being more electrostatically stable due to higher repulsion between the particles. Regarding the influence of the ambient pH on the in vitro release profile of 5-fluorouracil, comparing between liposomes and immunoliposomes, the authors concluded that the physiological pH (pH 7.4) did not interfere with drug release in contrast to the tumor tissue pH (pH 5.5), but, even so, at acidic pH, the amount of drug released from the immunoliposomes was only slightly less than that released from the regular liposomes.

Skin penetration studies were also carried out, in excised porcine ear skin, comparing liposomes, immunoliposomes, and a 5-fluorouracil-loaded solution delivered by iontophoresis. Iontophoresis application showed a greater increase in the amount of drug in the viable epidermis for the solution and a smaller increase for the liposomes when compared to passive application. The iontophoresis application of immunoliposomes increased 5-fluorouracil penetration within the viable epidermis, which is the usual localization of most skin cancer cells, which may be a result of the presence of EGFR in the viable epidermis.

Cellular uptake studies of liposomes and immunoliposomes, without 5-fluorouracil encapsulation but with a lipid fluorescent dye (DiO), were performed using the B16F10 (EGFR-negative) and A431 (EGFR-positive) cancer cell lines (Figure 8B). According to these in vitro studies, the cell uptake of immunoliposomes by A431 cells was higher than that of control liposomes due to endocytosis mediated by antigen–antibody interaction. In contrast, B16F10 cells showed minimal cellular uptake for both formulations. Thus, the higher uptake of immunoliposomes may be relevant in the treatment of A431 tumors, leading to a higher concentration of chemotherapeutic agents in tumor cells. 

To additionally analyze the effect of the route of administration on the results of the developed nanometric platforms, the authors used xenograft animal models, consisting of immunosuppressed mice to prevent cell rejection, with human tumor cells. These in vivo studies (Figure 8C), in immunodeficient Swiss nude mice, showed that iontophoresis followed by the topical administration of the developed immunoliposomes proved to be more effective than all other formulations and administration routes. In conclusion, this study showed that 5-fluorouracil-loaded immunoliposomes delivered by topical iontophoresis are an effective treatment for SCC, when compared to conventional therapies. This is one more proof that nanotechnology can overcome the limitations of chemotherapy by reducing off-target effects and the minimum drug concentration necessary for therapeutic effect.

### 2.5. 5-Fluorouracil and Resveratrol

Novel vesicular systems, including ultradeformable liposomes (ULs), have been investigated and demonstrated enhanced drug permeation rates following topical application. Edge activators, namely, SC, demonstrated in a study by Cosco et al. [41], to positively destabilize the lipid bilayer of the liposomal nanovesicles, modifying the interfacial tension and inducing deformability to the bilayer (Figure 9D). In this study, resveratrol- and 5-fluorouracil-loaded ULs were investigated for their potential use in the treatment of non-melanoma skin cancer, with a particular focus on SCC, including conditions such as Bowen’s disease, actinic keratosis, and keratoacanthoma. This combination of compounds was selected on the basis of their ability to promote the apoptosis of cancer cells in a synergistic manner, with resveratrol enhancing the effects of 5-fluorouracil, regardless of the presence or absence of p53. Resveratrol is a polyphenolic active compound produced by more than 70 plant species, including the ones that originate grapes, blueberries, and peanuts. In addition to its cardioprotective effects, resveratrol possesses a number of other noteworthy bioactive properties, including anti-inflammatory, antioxidant, and anticancer activities. In light of these additional properties, resveratrol may prove to be a valuable adjunct therapy to conventional chemotherapy regimens [77]. As a chemopreventive agent, resveratrol exerts its effects at various stages of carcinogenesis, including initiation, promotion, progression and metastasis [78]. Additionally, 5-fluorouracil, a well-known chemotherapeutic, has been demonstrated to exert a significant antitumor effect on skin cancer. This fluoropyrimidine analogue has the capacity to interfere with the synthesis of DNA, effectively blocking the conversion of deoxyuridylic acid to thymidylic acid [79].

The preparation of ULs entailed the dissolution of phospholipon 90G^®^ and SC in ethanol, within a glass vial. The formulation was prepared using the TFH method, with a water/ethanol solution, and subsequently subjected to sonication. The multi-drug ultradeformable liposomal formulations (MD-ULs) were obtained with the co-encapsulation of both resveratrol and 5-fluorouracil, which were respectively dissolved in their suitable environments. The drug-loaded UL pellets were washed with phosphate-buffered saline (PBS) and subjected to centrifugation, with the resulting supernatants stored for subsequent analysis. The resulting pellets were then redispersed in PBS buffer, thus preparing them for additional characterization

The mean particle size was determined by DLS, while the entrapment efficiency was evaluated by HPLC. Additionally, other parameters, including PDI and ZP, were analyzed in relation to varying drug concentrations. The EE was determined by the direct method. Furthermore, the authors evaluated the drug release profile by dialysis. The co-encapsulation of both drugs resulted in a lower average size of the UL than that observed for single drug liposomes up to a drug content of 3% (*w*/*w*). Thus, the combination of the two drugs did not result in any significant alteration in the size of the ULs in comparison to the single drug systems. These findings may be attributed to the lipophilic properties of resveratrol, which enable its localization within the bilayer of the ULs, and subsequent interaction with the bilayer components, thereby influencing their arrangement based on the amount of drug content. Regarding the ZP, which was observed to be within the range of 25 to 30 mV for all formulations, it was concluded that neither of the drugs under investigation exerted any significant influence on the surface charge of the liposome bilayers. The findings demonstrated that the co-encapsulation of both drugs also resulted in an increased entrapment of 5-fluorouracil into the ULs, which may be dependent on the localization of resveratrol within the liposome. The results of the HPLC analysis demonstrated that the constituents of the vesicles did not interfere with the encapsulation process, with an EE greater than 96% in all cases. The multi-drug ULs demonstrated the capacity to encapsulate a substantial quantity of resveratrol and a greater amount of 5-fluorouracil than the formulation containing this compound as a single drug. Therefore, the formulation comprising ULs loaded with both drugs at a concentration of 5% (*w*/*w*) was deemed as the most promising for further investigation. 

With regard to the drug release evaluation of the developed dual-loaded ULs, the hydrophilic drug 5-fluorouracil was released at a slower rate than when encapsulated solo, with around 80% being released after 24 h. On the contrary, resveratrol demonstrated a similar release profile both when in combination with 5-fluorouracil and when not, with an almost identical release amount after 48 h (around 90%). Consequently, the release of 5-fluorouracil was affected by the co-encapsulation of both chemotherapeutics into ULs, where resveratrol modulated the liposome bilayer fluidity and slightly blocked the overall delivery of 5-fluorouracil. Nevertheless, both drugs were still released successfully, to a high extent. Furthermore, regarding the permeation of the drugs through the *stratum corneum* and viable epidermis of the human skin, the encapsulation of 5-fluorouracil and resveratrol in ULs demonstrated a greater efficacy than that of the free drugs. Additionally, to assess the enhanced delivery of UL-encapsulated drugs to the dermis, a human full-thickness skin model was employed in the experiments. The amount of 5-fluorouracil and resveratrol retained in the dermis was, respectively, 97% and 93% of the amount applied to the skin. Thus, the ULs demonstrated the capacity to markedly enhance the dermal concentration of both drugs in comparison to their free counterparts. These findings may have implications for the development of topical dermatological formulations, as they illustrate the potential of ULs as a depot for drug delivery to the skin.

Moreover, the cytotoxic effect of the developed ULs was evaluated through a tetrazolium bromide (MTT) assay. Colo-38 and SK-MEL-28 cells treated with the ULs at concentrations ranging from 1 to 10 μM exhibited a reasonable cytotoxicity after 24 h of incubation (Figure 9E), and, following a 48-h incubation period, they exhibited an even more substantial antiproliferative activity (Figure 9F), even at the lowest tested drug concentrations. In comparison with the ULs, the free drug formulations yielded equivalent results after 24 h of incubation, while a lower level of cytotoxicity was observed after 48 h. These results were further confirmed by confocal laser scanning microscopy, where the accumulation of fluorescent ULs was evident within the cancer cells (Figure 9A–C).

The authors also verified the synergistic action of 5-fluorouracil and resveratrol by blocking the cell cycle in the G1/S stage and promoting 5-fluorouracil-induced apoptosis. Indeed, the physical mixture of resveratrol and 5-fluorouracil resulted in a notable increase in DNA strand interruption compared to isolated resveratrol. The apoptosis of the skin cancer cells was also analyzed through a spectrofluorimetric terminal deoxynucleotidyl transferase-mediated deoxyuridine triphosphate nick-end labeling (TUNEL) assay, which demonstrated that the treatment with the developed ULs induced a higher degree of DNA fragmentation, accompanied by an increase in green fluorescence intensity. The measurement of caspase-3 activity also indicated that the treatment with the liposomal system exhibited the highest proteolytic activity.

Therefore, this study demonstrated that the co-encapsulation of 5-fluorouracil and resveratrol in a multi-drug carrier such as ULs enhanced their antiproliferative activity on skin cancer cells, with the co-encapsulation possibly enhancing the efficacy of the two drugs. It may, therefore, be reasonably predicted that this formulation could have a potential clinical application.

### 2.6. Quercetin and Resveratrol

The co-delivery of quercetin and resveratrol, previously encapsulated in liposomes, was the subject of the study by Caddeo et al. [16] (Figure 10A). In addition to their relevant antioxidant effects, these polyphenolic compounds have other pharmaceutical benefits, such as being of natural origin (from vegetables and fruits) and having a high safety profile (with a Generally Recognized as Safe status) and proven anti-inflammatory and anticarcinogenic activities. Nevertheless, the poor water solubility of quercetin and resveratrol, as well as their physicochemical instability with changes in pH, temperature, and light, hamper their potential pharmaceutical benefits. Thus, they were incorporated into liposomes, and, in addition to the bioactive compounds, oleic acid was added to the dual-loaded vesicles, increasing their bilayer fluidity. Tris buffer was also added, as well as soy phospholipid (lipoid S75), composed by soybean lecithin with phosphatidylcholine, phosphatidylethanolamine, lysophosphatidylcholine, triglycerides, fatty acids, and tocopherol. The liposomes were prepared by TFH followed by sonication. 

The authors evaluated the average diameter, PDI, and ZP of the nanosystems by dynamic and electrophoretic light scattering and EE by HPLC. The average diameter results indicated small-sized vesicles, the PDI values revealed a monodisperse system, and the negative values of ZP were influenced by oleic acid and lipoid S75’s negative charge. Cryogenic transmission electron microscopy (cryo-TEM) analysis revealed small and spherical vesicles, mainly unilamellar, which was corroborated by small-angle X-ray scattering (SAXS) measurements. The EE was similar between quercetin and resveratrol and did not decrease over 2 months of storage. Turbiscan^®^ technology was also used to identify destabilizations (Figure 10C), and the co-loaded liposomes demonstrated good stability regarding changes in particle size and tendency to aggregate.

To measure the antioxidant activity of the bioactive compounds, the authors studied their scavenging ability of the free radical 2,2-diphenyl-1-picrylhydrazyl (DPPH). With the capture of the odd electron, the solution starts to discolor, demonstrating a certain scavenging efficiency. As a result, DPPH was nearly entirely inhibited by the co-loaded liposome, showing that the antioxidant activity of the polyphenols was maintained after encapsulation. The in vitro uptake and cytotoxicity of quercetin and resveratrol, as well as their action against ROS, were studied in human fibroblasts (Figure 10B). The results led to the conclusion that the developed co-loaded liposomes had a reduced cytotoxicity in fibroblasts compared to the isolated polyphenol vesicles. The co-integration of quercetin and resveratrol into the liposomes also led to a higher cellular uptake when compared to individual compounds (formulated into liposomes or not), also showing a greater ability of ROS scavenging in fibroblasts.

To evaluate the in vivo efficacy and safety of the developed liposomal formulation, the authors chose a mouse model for studying of myeloperoxidase (MPO) activity. In this study, 12-O-tetradecanoylphorbol 13-acetate (TPA) was used for tumor and wound promotion. Macroscopic observations of the skin only exposed to TPA revealed thickening, dryness, and an extensive crusting lesion. On the other hand, the administration of empty liposomes induced the formation of a protective film, promoting the hydration of the *stratum corneum*, and the treatment with the dual-loaded liposomes revealed an almost entirely re-established skin integrity and a completely healed wound. MPO and edema were also inhibited by the combination of quercetin and resveratrol within the liposomes. It is worth noting that empty liposomes also demonstrated an important role beneficial effect in edema and MPO tests as a result of the phospholipid antioxidant effect and the countering of inflammatory processes. Thus, the topical administration of the developed dual-loaded liposomes demonstrated an improvement of the tissue damage with a significant decrease in the chemical-induced migration of leukocytes and edema, which are important markers of the cancer inflammatory cascade.

### 2.7. Paclitaxel and DNA

Liu et al. [1] evaluated the performance of dual targeting hyaluronic acid (HA) and folate (FA)-modified liposomes (HA/FA/PPD) co-loaded with paclitaxel (PTX) and DNA (Figure 11A). Despite PTX’s general use for cancer treatment, including in melanoma, monotherapy has proven to have severe limitations. In this context, to reach a synergistic therapeutic effect, the combination of PTX with gene therapy was studied. However, the non-specific distribution through blood circulation and the potential immunological system’s response was identified as a limitation for combined PTX gene treatment application. To overcome these issues, nanocarrier co-encapsulation was selected as a solution to co-deliver DNA and PTX in cationic multilamellar liposomes.

Dual targeting liposomes were prepared by the TFH method. Empty liposomes were prepared with DOPE, chloroform, and egg L-α-phosphatidylcholine. HA/FA/PPD preparation included polyethylenimine (PEI) linkage to DNA. The polymer PEI was condensed with DNA, resulting in a final molar ratio of N/P = 10/1 (PEI nitrogen/DNA phosphate), with the complex localization being in the cationic core. PEI was meant to have an important role in DNA protection against DNase degradation and also enhance transfection efficiency. PXT and the cationic complex were co-loaded in the FA-modified liposomes, forming FA-modified liposomes (FA/PPD). Then, FA/PPD were added to a HA anionic solution by electrostatic attraction, obtaining HA/FA/PPD. To produce an optimal formulation, the defined mass ratio of HA and FA/PPD was 3:1. It is worth noting that, due to folate receptor overexpression in melanoma cells, FA was meant to allow specifical binding to tumor cells, inducing selective drug/gene delivery. However, FA/PPD might interact with serum complexes, leading to further a phagocytosis of the aggregates and, consequently, the release and degradation of PTX/DNA. For this reason, HA was added, a negatively charged polysaccharide intended to overcome the cationically charged FA/PPD aggregation by coating the surface of FA/PPD through electrostatic attraction. This resulted in an anionic and biomimetic layer. Transposing to an in vivo context, HA/FA/PPD was therefore meant to bind to overexpressed CD44 in cancer cells, and the HA layer was subsequently meant to be degraded by enzymes, exposing FA and targeting cancer cells once again.

The co-loaded biomimetic liposomes were characterized by particle size, PDI, and ZP determination (Figure 11B), and laser light scattering was used to measure them. The PTX in vitro release profile from the developed HA/FA/PPD was evaluated by dynamic dialysis technique, resulting in the sustained release of PTX from the nanosystem, possibly due to a slower degradation and erosion of the vesicle elements (Figure 11C). Furthermore, the developed liposomal system demonstrated an enhanced concentration-dependent cytotoxicity in B16 cells (melanoma) (Figure 11D). The enhanced cytotoxicity could be possibly attributed to the successful delivery of the drug molecules in the co-loaded liposomes. Additionally, to confirm the protective effect of HA on the liposomal surface, the authors evaluated the stability of HA/FA/PPD, and FA/PPD as well, by measuring the DNA protection ability against DNase I cleavage. HA/FA/PPD demonstrated a significantly enhanced effect on nanoparticle protection from aggregation, when compared to co-loaded FA/PPD, in the presence of DNase I degradation and plasma. Furthermore, the HA/FA/PPD liposomes demonstrated improved cellular uptake (Figure 11E) and transfection efficiency, when compared to PPD and FA/PPD liposomal systems, showing an increased therapeutic efficiency in vitro. Thus, as well as the protective action of the HA layer, this polysaccharide was also proven to be able to enhance cell internalization via CD44-mediated mechanism. The delivery effectiveness of PTX and DNA from HA/FA/PPD in the same cancer cells was confirmed by flow cytometry and fluorescence microscope. In fact, HA/FA/PPD led to a significantly higher co-deliver of PTX and plasmid DNA into the same tumor cell, when compared to FA/PPD, demonstrating the enhanced co-delivery efficiency promoted by HA. This study shows that combining chemotherapeutic drugs and nucleic acids can be a promising approach in cancer treatment.

### 2.8. Curcumin and STAT3 siRNA

In two studies performed by Jose et al. [80,81], curcumin was encapsulated in deformable cationic liposomes and, subsequently, complexed with the signal transducer and activator of transcription-targeted small interfering RNA (STAT3 siRNA). Curcumin, a compound obtained from the roots of *Curcuma longa*, is a lipophilic substance with anti-inflammatory, antioxidant, and anticancer properties, and it has been proven to have chemopreventive and chemotherapeutic activity in skin cancer models [81,82,83]. In fact, curcumin interferes with peroxisome proliferator-activated receptor gamma (PPARγ), STAT3, MAPK, p53, and nuclear factor kappa B (NFκB) pathways, inducing the apoptosis of cancer cells [82,84]. On the other hand, STAT3 is an oncogenic transcription factor that is activated in cancer [85,86]. It is involved in cell growth, invasion, metastasis, and the inhibition of apoptosis [87], and it activates vascular epidermal growth factor, survivin, and cyclin D and B-cell lymphoma-extra-large (Bcl-xl) family signaling molecules [88]. Although there is no report of specific small molecule inhibitors of STAT3, a gene-silencing strategy was chosen to inhibit the expression of the STAT3 protein. The gene-silencing agent, siRNA STAT3, can bind to a specific sequence of the target mRNA and inhibit the translation of the STAT3 protein. Therefore, STAT3 represents a potential target for skin cancer treatment that can be silenced using small interfering RNA, siRNA. 

Nevertheless, curcumin has reduced water solubility and low permeability across the skin barrier [89,90]. To overcome these limitations, there are some strategies to increase the solubility of curcumin, including cyclodextrin complexation, and encapsulation in micelles or liposomes [91,92]. Liposomal vesicles are good vehicles for curcumin encapsulation and skin penetration [93] and retain flexibility to penetrate the skin pores, which increases their stability, compared to micelles [80,94]. The authors also reported the limitations of siRNA as an isolated compound. The high molecular weight of siRNA and its negative charge limit its transport through negatively charged skin [95]. The complexation of siRNA into nanosized particles, more specifically cationic liposomes, was meant to counteract the rapid degradation of isolated siRNA, overcoming its in vivo instability, and fight against its poor cellular uptake [80,96].

Cationic liposomes were prepared by TFH, using DOTAP, DOPE, C6 ceramide, SC, and ethanol, with chloroform as the organic solvent of choice, which was removed under pressure, and 20 mM HEPES buffer (pH 7.4) to hydrate the dried lipid film. The liposome suspension was then sonicated and extruded through a 100 nm pore size polycarbonate membrane to increase homogeneity and reduce particle size. Sufficient flexibility to penetrate the *stratum corneum* was ensured by using SC as an edge activator. Anodal iontophoresis was used to enhance the skin penetration of the positively charged liposome-siRNA complexes.

Jose et al. [80] investigated the average particle size, PDI, and ZP of the developed liposomes by the Dynamic Light Scattering (DLS) technique, comparing liposome-siRNA complexes, curcumin loaded liposomes, and curcumin loaded liposome-siRNA complexes, stored at 2–8 °C for 90 days. They also determined the entrapment efficiency of curcumin retained at different storage times in curcumin-loaded liposomes, stored at 2–8 °C for 90 days, as well as the drug release profile using a Franz diffusion cell apparatus. The authors used the indirect method to determine the EE, after separating the free curcumin from the liposome suspension by centrifugation. The total curcumin concentration was obtained by lysing the liposomes of the suspension. The concentration of entrapped curcumin was then determined by high performance liquid chromatography (HPLC). The percentage of curcumin retained in the liposome-siRNA complex decreased from 100% to 92% after 1 month. To optimize the maximum EE, the authors considered a 10:1 formulation (lipid to curcumin ratio) in their experiments. Analysis showed that there was no significant change in ZP and particle size up to 30 days of storage. In the other study performed by Jose et al. [81], the stability parameters were also investigated and showed that the encapsulation of curcumin did not affect with the average particle size or ZP. The positively charged liposomes allowed the complexation of STAT3 siRNA due to the presence of DOTAP.

Skin permeation was also studied, using excised porcine ear skin, after passive and anodal iontophoretic application. Tape stripping allowed for the analysis of the amount of curcumin retained in the *stratum corneum* and viable skin after treatment with free and liposomal curcumin. The amount of curcumin retained in the *stratum corneum* was comparable 48 h after treatment with liposomal curcumin and free curcumin. However, anodal iontophoretic application significantly enhanced the *stratum corneum* penetration of liposomal curcumin, compared to passive application, and resulted in a 5-fold greater deposition of curcumin in viable skin. Similar results were obtained for curcumin-loaded cationic liposome–Cy3 siRNA complexes: after iontophoresis application, the curcumin-loaded liposome–Cy3 siRNA complex was able to reach the viable epidermis. Therefore, the application of anodal iontophoresis improved the skin permeation of curcumin-loaded liposome–siRNA complexes to reach a target depth of up to 100 μm within the skin [80].

The lipophilic nature of curcumin allowed it to pass more easily through the liposomal phospholipid bilayer. The elasticity of deformable liposomes provided them resistance to mechanical stress, allowing the liposomes to deform and to cross the channels within the cells of the *stratum corneum*, the epidermis, and to reach the dermis. The elasticity value of liposomes with SC (20.2 ± 1.5 mg s^−1^ cm^−2^) was four times higher than that of liposomes without SC (4.6 ± 0.5 mg s^−1^ cm^−2^) [65,80]. 

Cell uptake was also investigated in human epidermoid carcinoma cells (A431). Curcumin-loaded liposomes complexed with siRNA were labeled with a fluorescent dye, Cyanine-3 (Cy3). Flow cytometry and geometric mean fluorescence methods indicated an increased cell uptake of the cationic liposome–Cy3 siRNA complex within the time required for successful therapeutic activity. The cell uptake of the curcumin-loaded liposomes and curcumin-loaded liposome–Cy3 siRNA complex was also investigated using two endocytosis uptake inhibitors. Methyl-β-cyclodextrin and chlorpromazine were used to selectively inhibit of caveolae-mediated and clathrin-mediated endocytosis, respectively. The results showed a greater inhibition of cell uptake after pre-treatment with chlorpromazine hydrochloride compared with methyl-β-cyclodextrin [80].

Additionally, cell viability studies were conducted in A431 cells utilizing varying concentrations of the developed liposome–siRNA complexes (0.25 nM, 0.5 nM, and 1.0 nM). The findings demonstrated that the 0.5 nM siRNA resulted in a considerably greater cell growth inhibition when compared to the 0.25 nM concentration. Moreover, no significant difference was observed in the inhibitory effect of the 1.0 nM siRNA complex when compared with 0.5 nM siRNA complex. Consequently, 0.5 nM siRNA was selected as the concentration for the co-delivery of curcumin and STAT3 siRNA using liposomes, containing 250 μM of encapsulated curcumin, which demonstrated the greatest growth inhibition. Moreover, the occurrence of late apoptosis was found to be elevated following treatment with the co-delivery system comprising both curcumin and STAT3 siRNA in comparison to the administration of curcumin-loaded liposomes and the liposome–STAT3 siRNA complex, which exhibited lower levels of late apoptosis. Furthermore, in studies examining the suppression of STAT3 protein, free curcumin did not demonstrate any significant inhibitory effect. However, curcumin-loaded cationic liposomes were observed to suppress STAT3 protein expression. The addition of STAT3 siRNA to the curcumin-loaded cationic liposome complex resulted in a further reduction in STAT3 protein expression. Based on these findings, it can be claimed that the curcumin-loaded liposome–STAT3 siRNA complex may offer an effective approach to inhibiting skin cancer cell growth [80].

The tumor volume change was also evaluated in vivo [81] in an efficacy study performed in a mouse model. The tumor volume was calculated by measuring the length and breadth of the tumor at regular intervals, using a digital vernier caliper. The dual delivery of curcumin and STAT3 siRNA liposomes was found to significantly inhibit tumor development, when compared with liposomal curcumin or STAT3 siRNA alone. The results of an immunohistochemical analysis of STAT3 protein expression within the tumors indicated that the greatest fluorescence intensity was observed in the cryosections of the untreated control, the passive application of the liposome-siRNA complex, and the iontophoretic application of the liposome-scrambled siRNA complex. Conversely, the lowest fluorescence intensity was observed in the intratumoral injection and iontophoretic application of the curcumin-loaded liposome–STAT3 siRNA complex. In conclusion, the iontophoretic administration of the curcumin-loaded liposome–siRNA complex demonstrated comparable efficacy in STAT3 protein suppression and tumor progression inhibition to that observed following intratumoral administration.

### 2.9. Aurora-A Inhibitor XY-4 and Bcl-xl siRNA

Cell cycle kinase inhibitors combined with other cancer treatment agents have demonstrated improved effects. In this context, Duan et al. [12] studied the co-delivery within cationic liposomes of Aurora-A kinase inhibitor XY-4 and Bcl-xl targeted siRNA, through injectable administration. Aurora-A kinase inhibitor XY-4 is a cell cycle kinase inhibitor that specifically targets Aurora subtype A. This chemotherapeutic agent acts in G2/M cell cycle phase by blocking it and inhibiting cell proliferation. Aurora-A kinase is located in the spindle poles and centrosomes of the cell and can recruit cyclin B1-CDK1 complex to induce mitosis. Drug resistance may be related with the amplification of the oncogenic Aurora-A kinase. XY-4 has a pyrazolo[3,4-b] pyridine scaffold structure, interacting with Aurora-A kinases. On the other hand, the Bcl-xl protein is an antiapoptotic agent that prevents mitochondrial content release. It is worth noting that cytochrome c, an element of mitochondrial content, can lead to the activation of caspase and subsequent apoptosis. Bcl-xl-targeted siRNA strengthens the therapeutic effect of Aurora-A kinase inhibitor XY-4 by silencing Bcl-xl and, consequently, inducing apoptosis.

The preparation of XY-4-loaded liposomes was based on the TFH method, using DOTAP, cholesterol, and Aurora-A kinase inhibitor XY-4. At the end of this method, unloaded XY-4 was removed. Furthermore, Bcl-xl-targeting siRNA was added to previously prepared XY-4 liposomes by incubation. XY-4 was integrated in the liposome core, while Bcl-xl-targeting siRNA was bound to positively charged DOTAP.

The liposomes were characterized by particle size, ZP, PDI, EE (direct method), and drug release. The results indicated a reduced PDI value, lower than 0.3, resulting in monodispersed liposomes, with an extremely limited particle size distribution. XY-4-loaded liposomes’ EE and drug loading were also determined, resulting in 84.6% and 4.76%, respectively. The results also confirmed the high stability of the vesicles. The in vitro drug release was performed through the dialysis method and allowed for the determination of the XY-4-loaded liposome release profile, with results showing a considerable slower release rate when compared to free XY-4, thus demonstrating a sustained drug delivery, contributing to a prolonged therapeutic efficacy and safety. Transfection efficiency assays were also performed, demonstrating a high siRNA transfection and, consequently, efficient siRNA delivery through the liposomes.

Cell uptake studies were conducted as well, in B16 mice melanoma cells, and performed with a fluorescent compound, coumarin-6, as a model drug. Cationic liposomes were preferably captured by B16 melanoma cells, compared to neutral liposomes, showing more efficiency and improvement in drug cellular uptake. This information suggested that the delivery of XL-4 was facilitated by cationic liposomes. The combined liposome therapy was also analyzed to further enhance the therapeutic effect of XY-4. In fact, flow cytometry showed that XY-4 significantly inhibited cell cycle. The cellular proliferation was evaluated by the MTT assay, showing that XY-4 could inhibit B16 cell proliferation. The therapeutic combination also demonstrated an effective inhibition of B16 cell proliferation, suggesting a promising therapeutic effect of combined Bcl-xl siRNA and XY-4. These effects were also proven to occur via mitochondrial apoptosis pathway. Additionally, Western blotting results revealed that the co-loaded formulation group also increased the cytochrome c levels in melanoma cells, as well as decreased anti-apoptotic proteins Bcl-xl expression. The combination group treatment also induced a higher expression of cleaved caspase-3 and cleaved caspase-9 and a lower expression of caspase-9.

B16 melanoma cells were injected into the right flanks of female C57 mice, for the performance of in vivo assays. After the intratumoral injection of the co-loaded formulation, melanoma cell growth in the xenograft models was proven to be efficiently inhibited. When compared to isolated drug formulations, the co-administration of Aurora-A kinase inhibitor XY-4 and Bcl-xl siRNA promoted a statistically significant tumor weight reduction. The results also proved that the developed co-loaded formulation was potentially safe due to the absence of obvious pathological modifications in the animals after administration, such as ruffling of fur, weight loss, and behavior changes. Thus, in vivo studies demonstrated high safety and strong therapeutic efficacy. Overall, the results of this study showed that Aurora-A kinase inhibitor XY-4 and Bcl-xl-targeted siRNA co-administration might be a promising treatment for melanoma, depicting beneficial synergistic effects.

### 2.10. 1-Methyl-Tryptophan and Cytosine–Phosphate–Guanosine Anionic Peptide

Su et al. [97] designed a cationic polymer–lipid nanocarrier to deliver water-soluble cancer vaccines, composed of anionic antigen epitope cytosine–phosphate–guanosine anionic peptide-modified epitope (AE/CpG), a toll-like receptor-9 agonist (TLR9), as well as 1-Methyl-tryptophan (1-MT), an indoleamine-2,3-dioxygenase (IDO) inhibitor. 1-MT was added to improve the immunogenicity of the antigens and inhibit the immune checkpoint. Dendritic cells play an important role in immunotherapy as antigen-presenting cells (APCs). APCs are responsible for the recognition of the tumor antigen and subsequent uptake and process, with simultaneous activation and maturation of APC. Afterwards, the antigen is presented to naïve T-cells, inducing cancer cell depletion by cytotoxic T lymphocyte stimulation. Due to peptide vaccines’ immunogenicity, adjuvants are combined with peptide vaccines to improve the immune response of T-cells. 1-MT can suppress IDO, an enzyme that inhibits T-cell activation by tryptophan level reduction. Thus, the objective of the proposed nanovesicle immunotherapy combination was to enhance efficacy by inhibiting immunosuppression and overcoming the low antigenicity of peptide vaccines.

1-MT was initially incorporated into cationic liposomes by TFH, which were subsequently complexed with AE/CpG, producing the cancer vaccine. D8 negatively charged peptide was conjugated with SIINFEKL, a specific melanoma epitope derived from antigen ovalbumin, able to bind to cationic liposomes. Particle size and PDI were measured by DLS. Encapsulation efficiency was determined by UV spectrophotometry. TEM determined a spherical liposomal morphology. AE/CpG encapsulation did not significantly affect the structure of liposomes but instead their size.

The developed liposomes increased the dendritic cells’ uptake of the vaccines in an efficient manner when compared to the free vaccine, which indicates that the delivery system did in fact increase endocytosis. Regarding the localization of the vaccines in the intracellular environment, liposomes encapsulated with AEs were found in the cytoplasm, while free vaccine AEs were mostly found in the lysosomes. The findings demonstrated that cancer antigens loaded in liposomes could evade lysosomal degradation, which allowed these antigens to be presented to CD8+ T-cells, stimulating the immune response against the tumor. The cross-presentation of antigens mediated by MHCI is a crucial process for CD8+ T lymphocyte activation, which are relevant cells in the antitumor immune response. Therefore, bone-marrow-derived dendritic cells were maturated by antigens from the liposome formulation and, subsequently, activate naïve T-cells inducing CTL responses. Co-loaded nanocarriers enhanced dendritic cell maturation as evidenced by the higher percentage of CD86+MHCI+ cells, when compared to AE/CpG or liposomal 1-MT. It also led to a robust cytotoxic T-lymphocyte reaction against in vitro B16-OVA tumor cells. B16 cell viability after liposomal formulation application was significantly reduced when compared to the free vaccine or liposomes loaded with 1-MT. These results demonstrated that liposomal peptide vaccines have the potential to elicit a robust antigen-specific CTL response against cancer cells due to dendritic cell activation by the developed formulation, leading to a greater presentation of antigens.

In vivo antitumor assays were performed as well in female C57BL/6 mice injected with B16-OVA melanoma cells. Treatment with the designed formulation successfully delayed the progression of the tumor. Combined immunotherapy led to a significant increased therapeutic efficiency in mice melanoma cells. The liposomal formulation loaded with tumor vaccines and 1-MT increased CD8+ and CD4+ lymphocytes in the tumor, proposing an improved T-cell melanoma immunity. The remarkable enhanced tumor infiltration of CD8+ T-cells and draining lymph nodes characterized a powerful T-cell response with specificity to the tumor site. The combined treatment biocompatibility was also determined with no obvious pathological abnormalities in the major mice organs, suggesting suitable biocompatibility in vivo. Thereby, cationic polymer–lipid liposomes loaded with tumor vaccines and 1-MT induced antitumor immunity that was T-cell-dependent, which improved cancer immunotherapy, demonstrating that the produced nanovesicles are a promising co-delivery system for tumor vaccines.

### 2.11. Bufalin and Anti-CD40 Antibody

In a study by Li et al. [19], the authors investigated the potential of co-delivery immunoliposomes containing bufalin and anti-CD40 to achieve enhanced melanoma treatment efficacy while reducing the occurrence of systemic adverse effects. Bufalin is a digoxin-like compound with antitumor effects derived from the toads *Bufo gargarizans* or *Bufo melanostictus*, particularly from their parotid glands and skin. This component has been demonstrated to inhibit cancer cell proliferation, induce cell cycle arrest, interfere with the immune response, and induce apoptosis. Nevertheless, the administration of bufalin has been associated with an increased toxicity, immunosuppression, drug resistance, and damage to healthy cells. On the other hand, CD40 is a tumor necrosis factor receptor that is expressed on the outer layer of a diversity of normal cells. The interaction between CD40 and its ligand delivers a signal to antigen-presenting cells, boosting their capacity to present antigens and promoting proinflammatory cytokines production. This in turn induces a cytotoxic antitumor response by T-cells. Furthermore, CD40 has the capacity to enhance the migration of leukocytes. Thus, the co-incorporation of both a monoclonal antibody and a chemotherapeutic agent was proposed to achieve a synergistic effect and reduce bufalin cytotoxicity and systemic adverse effects.

The liposomes were prepared using the TFH method, followed by sonication and extrusion of the formulation, with a 200 nm pore size membrane. The formulation was produced using cholesterol, L-α-phosphatidylcholine, 1,2-distearoyl-sn-glycero-3-phosphoethanolamine-N [maleimide (polyethylene glycol)-2000], 1,2-distearoyl-sn-glycero-3-phosphoethanolamine-N [methoxy (polyethylene glycol)-2000], and bufalin in a 20:55:5:5:15 molar ratio, respectively. Subsequently, anti-CD40 monoclonal antibodies were coupled to maleimide-functionalized liposomes on the unilamellar liposome surface, while bufalin was encapsulated within these vesicles. It is pertinent to highlight that PEG was employed to enhance the delivery, prolong circulation time, and improve local retention of the vesicles (and especially the drugs) in the tumor, while cholesterol was optimized in order to achieve a desirable EE and liposomal surface rigidity. Further changes beyond the optimized ratio may result in the precipitation of lipids and a reduction in the EE.

Particle size, PDI, and ZP were determined by DLS, morphology by TEM, and bufalin EE by HPLC. The EE was determined by the direct method, and the drug release was evaluated by dialysis. The formulation exhibited a narrow size distribution (PDI = 0.062 < 0.3) and spherical uniform shape. The combination of these factors with the low particle size and the relevant ZP were meant to permit the permeation of the liposome into the tumor microvasculature in a passive manner. Additionally, the co-loaded liposomes demonstrated higher cytotoxicity in B16 cells (melanoma) than free bufalin, and similar growth inhibition to bufalin liposomes, after a 24-h exposure. In fact, the IC50 of the co-loaded liposomes was found to be considerably lower than that of the free bufalin formulation but similar to that of the bufalin liposomes. These findings can be attributed to the enhanced cellular uptake of the liposomal formulations.

In vivo studies were performed in C57/BL6 female mice, with a prior subcutaneous injection of B16 cells. Then, the mouse model with B16 melanoma cells was intravenously injected with the developed co-loaded liposomes, resulting in a smaller tumor volume and tumor weight than the other treatment groups. This was consistent with the previously observed in the in vitro cytotoxicity assays, thus demonstrating that the therapeutic efficacy was improved with the dual-loaded liposomes. Hence, these studies revealed a synergistic antitumor effect between the two loaded compounds. Furthermore, animal body weight alterations were analyzed to predict the formulation’s systemic toxicity. In the case of the co-encapsulated liposome, the body weight variation was minimal in comparison to the anti-CD40 solution. This might be associated with the capacity of anti-CD40 to enter the systemic circulation, thereby eliciting a generalized inflammatory response. Consequently, inflammation may lead to the progressive loss of muscle mass. Thus, the dual-loaded liposomal system seemed to have led to a reduction in the incidence of systemic side effects, given the unapparent changes in animal body weight.

Furthermore, the developed delivery systems permitted simultaneous and long-lasting antigen delivery, with tumor apoptosis being evaluated by TUNEL and confirmed by Western blot analysis. The mechanism of action may involve the mitochondria-dependent apoptosis pathway, as evidenced by the elevated levels of caspases and cytochrome c that were identified in the Western blot analysis. Additionally, the co-loaded formulation demonstrated a significant reduction in certain serum cytokines levels, including tumor necrosis factor-α.

## 3. Comparative Discussion

A summary of the most relevant parameters of the analyzed studies, regarding dual-loaded liposomal systems for the treatment of skin cancer, including co-loaded drug molecules, particle size, PDI, zeta potential, encapsulation efficiency, and main biological findings, is present in Table 1.

To sum up, all the developed delivery systems were in the nanoscale range. The mean particle size of the delivery systems permitted the characterization of the liposomal formulations as either SUV or LUV. Consequently, the nanoscale dimensions of all nanosystems allowed them to reach deep layers of the skin, where tumors may be located. Moreover, the PDI values of the lipid-based carriers under analysis were all below 0.3, indicating the presence of homogeneous populations of nanocarriers with a specific size and a narrow size distribution. Most vesicles also had enhanced stability and low levels of liposome aggregation. A higher ZP absolute value (regardless of being positive or negative) indicated a greater electrostatic stability of the liposomal suspension, which helps to prevent liposome aggregation. Thus, there were some studies with demonstrated low repulsion interactions between liposomes, increasing the probability of aggregation, while other studies revealed good stability. Additionally, the EE of the studied therapeutic agents was in most cases higher than 85%. These values were regarded as being high, demonstrating an excellent degree of encapsulation efficiency.

Regarding formulation components, besides the drug molecules, the most commonly used in the analyzed studies were cholesterol, DOTAP, and PEG. In fact, the presence of cholesterol within the liposomal membranes increased their rigidity and stability. On the other hand, DOTAP, due to its positive charge, facilitated the interaction of the developed vesicles with negatively charged biological molecules, such as DNA and RNA. This made it a useful substance for the delivery of genetic material. It could also promote the fusion of liposomes with the cell membrane, which is an additional advantage. Furthermore, the conjugation of PEG to the nanosystems prevented opsonization, increasing the circulation time of the liposomes in the blood. It also prevented aggregation, increasing the colloidal stability and uniform particle sizes. As for nanosystem type, the most studied nanosystems were liposomes, immunoliposomes, and deformable cationic liposomes, when compared to UL, cationic liposomes, modified biomimetic liposomes, and cationic polymer–lipid hybrid nanovesicle-based liposomes.

With regard to the investigated therapeutic agents, the combined molecules were curcumin and STAT3 siRNA, 5-fluorouracil and resveratrol, 5-fluorouracil and cetuximab, quercetin and resveratrol, doxorubicin and hispolon, doxorubicin and ceramide, doxorubicin and celecoxib, Aurora-A inhibitor XY-4 and Bcl-xl siRNA, paclitaxel and DNA, 1-MT and CpG, and bufalin and anti-CD40 antibody. Therefore, the researchers emphasized the importance of polyphenols, flavonoids, pyrimidine analogues, anthracyclines, pyrazole derivates, taxanes, sphingolipids, tryptamines, non-steroidal anti-inflammatory drugs, cardiotonic steroids, nucleic acids, and antibodies in skin cancer treatment and their combined synergistic effects. Doxorubicin is among the most frequently used pharmaceutical agents for the treatment of a diverse spectrum of cancers due to its high effectiveness, which was reflected in it being included in more than one of the publications analyzed in this review.

Based on the analysis of the drugs’ therapeutic mechanisms of action, target specificity, and therapeutic synergy of the nanosystems, promising effective dual-loaded nanosystems of therapeutic molecules can be highlighted. One such combination is the deformable cationic liposome with curcumin and STAT3 siRNA [80,81], where the synergistic effect between an anti-inflammatory/antioxidant agent and an oncogenic transcription factor was particularly noteworthy, targeting inflammation and cancer progression. Furthermore, the deformation of the nanosystem improved penetration to the tumor site. Another notable example was the immunoliposome with the well-stablished chemotherapeutic agent 5-fluorouracil, and cetuximab [37], which was specifically targeted to EGFR and enhanced the skin penetration by iontophoresis. Another clear case was the immunoliposome with bufalin and anti-CD40 antibody [19], which involved a synergistic effect between a cytotoxic agent and immune stimulation, with effective and specific targeting, with reduced systemic side effects. These three examples of nanosystems combined different therapeutic approaches (chemotherapy, gene therapy, immunotherapy) and enhanced the delivery of therapeutic agents to their targets, thereby increasing their potential efficacy against skin cancer.

Despite the widespread pre-clinical investigation of co-delivery systems, there are some challenges in the potential transition to commercialization. The transferal of the developed formulations to clinical practice has been gradual. It is crucial to ensure ongoing collaboration and communication among experts throughout every phase of pharmaceutical development, from pre-clinical and clinical trials to toxicological evaluations [98]. Additionally, some liposomal preparation methods can make it difficult to produce the formulation on a large scale, as is the case of polycarbonate membrane extrusion [99]. Furthermore, maintaining stability over time is challenging to achieve, as demonstrated by the findings of Jose et al. [80], which indicate notable alterations in particle size, PDI, and ZP at the conclusion of the three-month study period.

## 4. Conclusions

Conventional cancer therapy is associated with limited therapeutic efficacy, severe adverse effects, and narrow therapeutic windows, which makes it impossible to increase the concentration of administered drug required to hinder cancer cell proliferation. On the other hand, small nanometric drug delivery systems have demonstrated higher permeability and enhanced retention in tumor tissues, enabling liposome accumulation at the therapeutic sites of action. The simultaneous delivery of several types of therapeutic agents, in dual-loaded liposomal systems, has been proven to induce synergistic antitumor activities in skin cancer, inducing higher cell cytotoxicity and improving global treatment outcomes, thus being an alternative strategy with high therapeutic efficacy and reduced side effects. In conclusion, this review of dual-loaded liposomal-based nanosystems for the treatment of skin cancer has revealed a dynamic and promising field of research, with notable advancements in drug delivery technology and therapeutic strategies. Advances in the use of deformable liposomes and microneedles have shown considerable potential for improving the efficacy and safety of skin cancer therapies. The observed diversity of therapeutic approaches included the combination of chemotherapeutic agents with gene therapy and the integration of immunotherapy. The therapeutic synergy provided by combining multiple agents within a single nanosystem was one of the main highlights, suggesting that these systems can interfere with different mechanisms of cancer progression more comprehensively. However, although the preliminary data seem to be promising, the long-term safety and efficacy of nanosystems still need to be confirmed through rigorous clinical studies. It is crucial that future research focuses on optimizing these systems, evaluating their efficacy in different types of skin cancer and overcoming the challenges related to large-scale industrial production and stability.

## Figures and Tables

**Figure 1 pharmaceutics-16-01200-f001:**
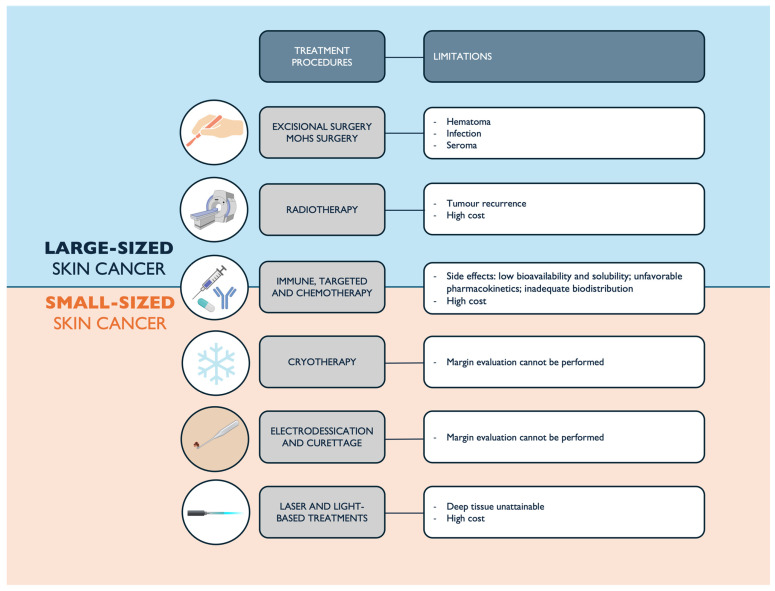
Representation of current skin cancer treatment procedures and the respective limitations (produced with BioRender https://www.biorender.com/).

**Figure 2 pharmaceutics-16-01200-f002:**
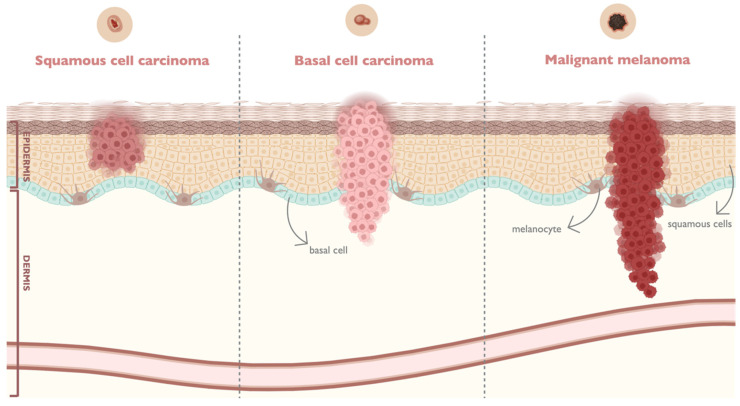
Schematic representation of melanoma and non-melanoma skin cancer (squamous cell carcinoma and basal cell carcinoma) (produced with BioRender https://www.biorender.com/).

**Figure 3 pharmaceutics-16-01200-f003:**
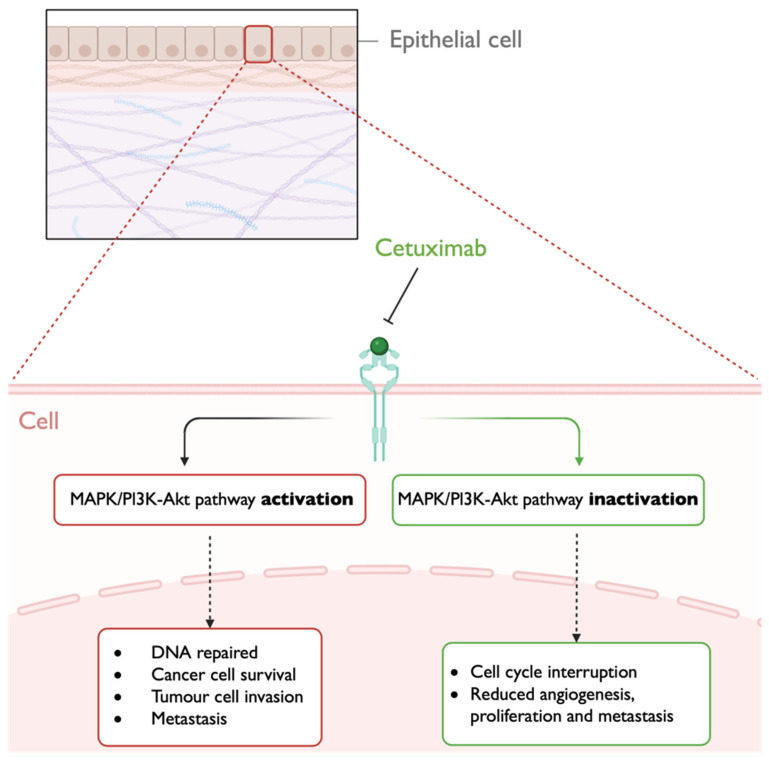
Targeting EGFR in skin cancer treatment. MAPK/PI3K-Akt pathway inactivation leads to the interruption of cell cycle, hampering tumor cell proliferation and metastasis (produced with BioRender https://www.biorender.com/).

**Figure 4 pharmaceutics-16-01200-f004:**
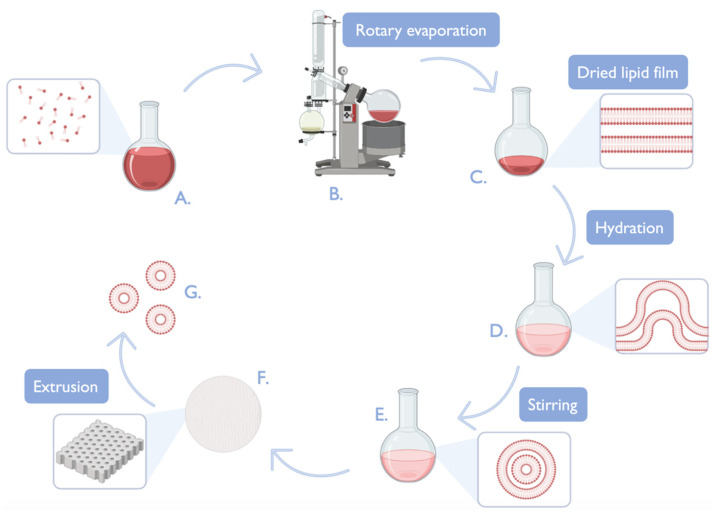
Thin-film hydration method, followed by extrusion. A—isolated phospholipids in organic solvent; B—rotary evaporation (organic solvent completely evaporated); C—dried lipid film (adjacent phospholipids with a linear structure); D—hydration (added water swells the dried lipid film) and agitation (vesicles form); E,F—extrusion of liposomal suspension through a polycarbonate filter with a specific pore size, allowing homogeneous-sized liposomes to cross the filter to the final formulation; G—homogeneous liposomes with similar particle size (produced with BioRender https://www.biorender.com/).

**Figure 5 pharmaceutics-16-01200-f005:**
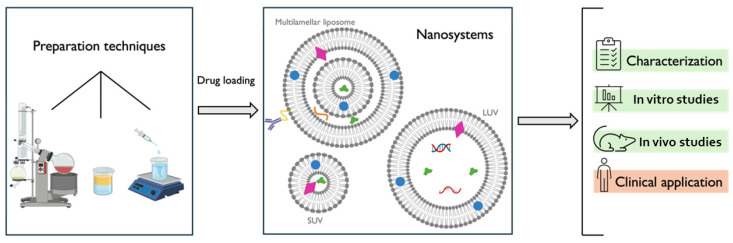
Schematic representation of the general procedure of development and characterization of liposomes and derived nanometric systems, from the preparation methods to the experimental studies, with particular relevance to the design of loaded nanocarriers with different sizes and numbers of bilayers. Structural characterization: small unilamellar vesicles (SUVs), large unilamellar vesicles (LUVs), and multilamellar liposome. Loaded nanosystem characterization: phospholipid (gray), hydrophobic drug molecule (light blue), hydrophilic drug molecule (green), amphiphilic molecule (pink), peptide (orange), antibody (purple), PEG (yellow), DNA (dark blue), and RNA (red) (produced with BioRender https://www.biorender.com/).

**Figure 6 pharmaceutics-16-01200-f006:**
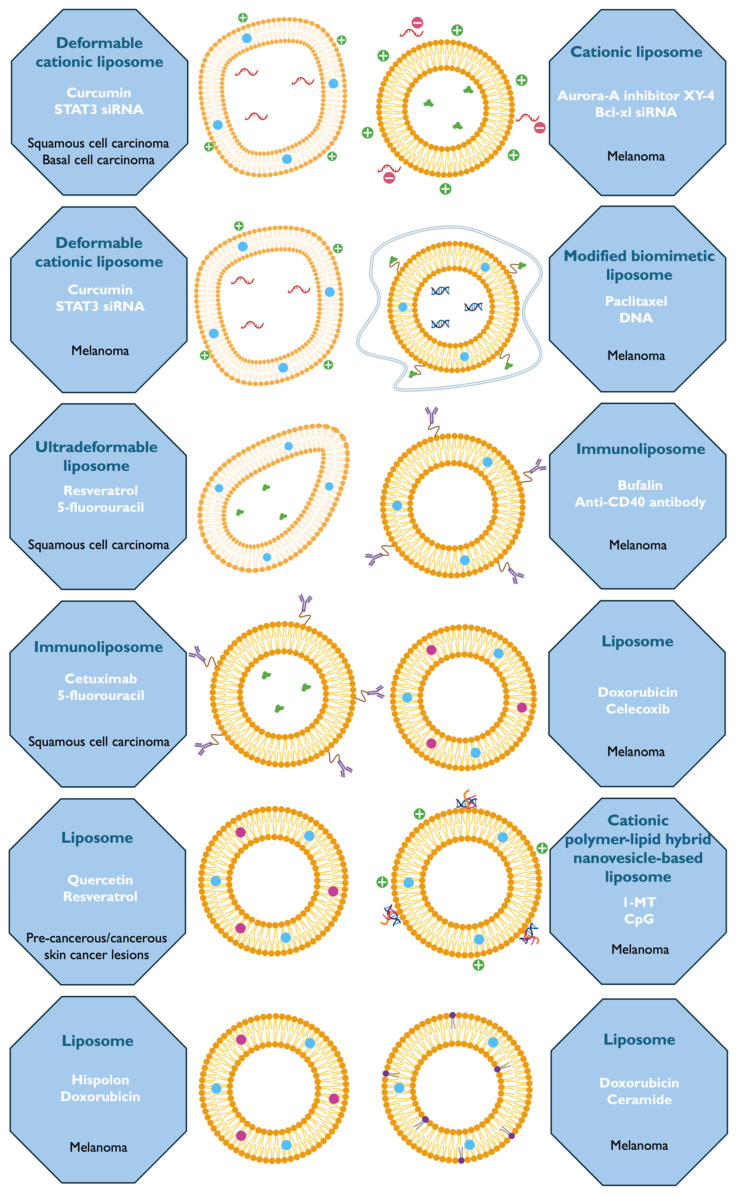
Diagrammatic representation overview of the type of liposomes, dual therapeutic agents, and skin cancer/lesions explored in each described study (produced with BioRender https://www.biorender.com/).

**Figure 7 pharmaceutics-16-01200-f007:**
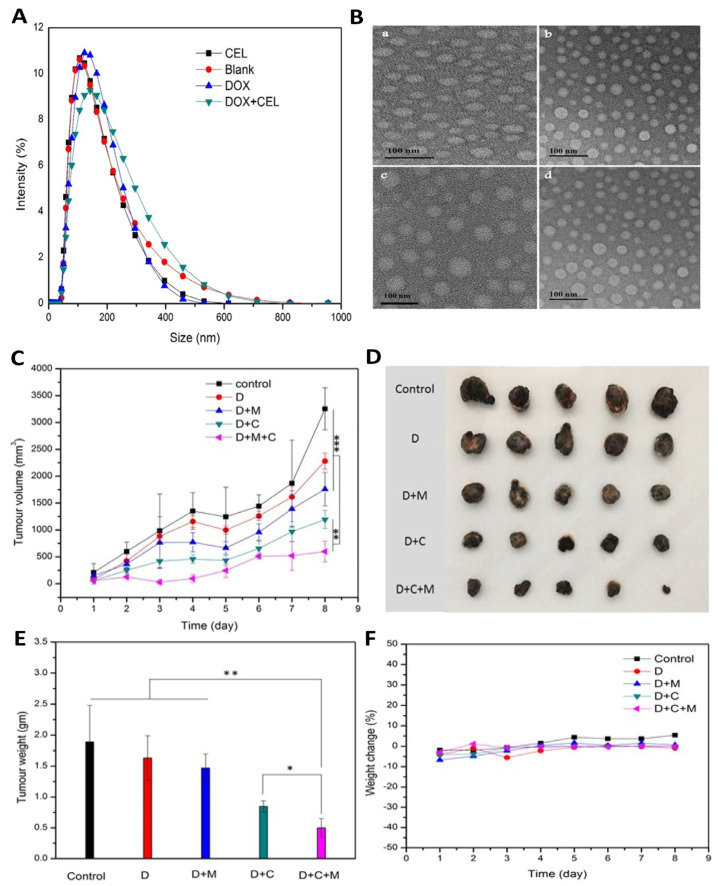
(**A**)—Particle size distribution of the developed liposomes; (**B**)—TEM micrographs of the developed formulations, namely, liposomal vehicles ((**a**) no drugs), CEL liposomes ((**b**) with celecoxib only), DOX liposomes ((**c**) with doxorubicin only), and DOX/CEL co-loaded liposomes ((**d**) with celecoxib and doxorubicin); (**C**)—tumor volume variation in B16 melanoma mice, after treatment with the developed liposomal formulations; (**D**)—tumor images in B16 melanoma mice, after treatment with the developed liposomal formulations; (**E**) tumor weight variation in B16 melanoma mice, after treatment with the developed liposomal formulations; (**F**)—mouse weight variation in B16 melanoma mice, after treatment with the developed liposomal formulations; * *p* < 0.05, ** *p* < 0.01, and *** *p* < 0.001; adapted from Ahmed et al. [43].

**Figure 8 pharmaceutics-16-01200-f008:**
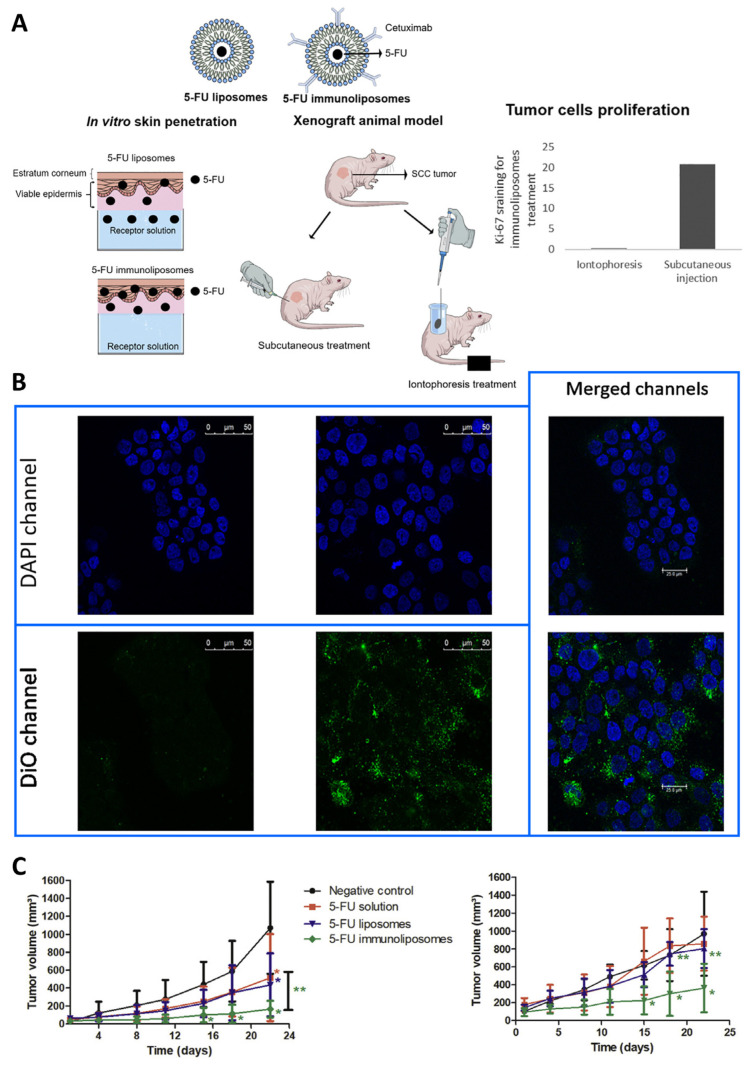
(**A**)—Schematic representation of the developed liposomal systems, along with the main assays conducted in this study; (**B**)—Confocal images of skin cancer cells (A431 cell line) 24 h after treatment with the developed liposomes and immunoliposomes, labeled with DiO, with cell nuclei having been labeled with DAPI; (**C**)—Tumor volume variation after treatment with the developed liposomal formulations, administered after topical iontophoresis (image on the left) or subcutaneously (image on the right); * *p* < 0.05 vs. PBS, and ** *p* < 0.05 5-fluorouracil solution vs immunoliposomes; adapted from Petrilli et al. [37].

**Figure 9 pharmaceutics-16-01200-f009:**
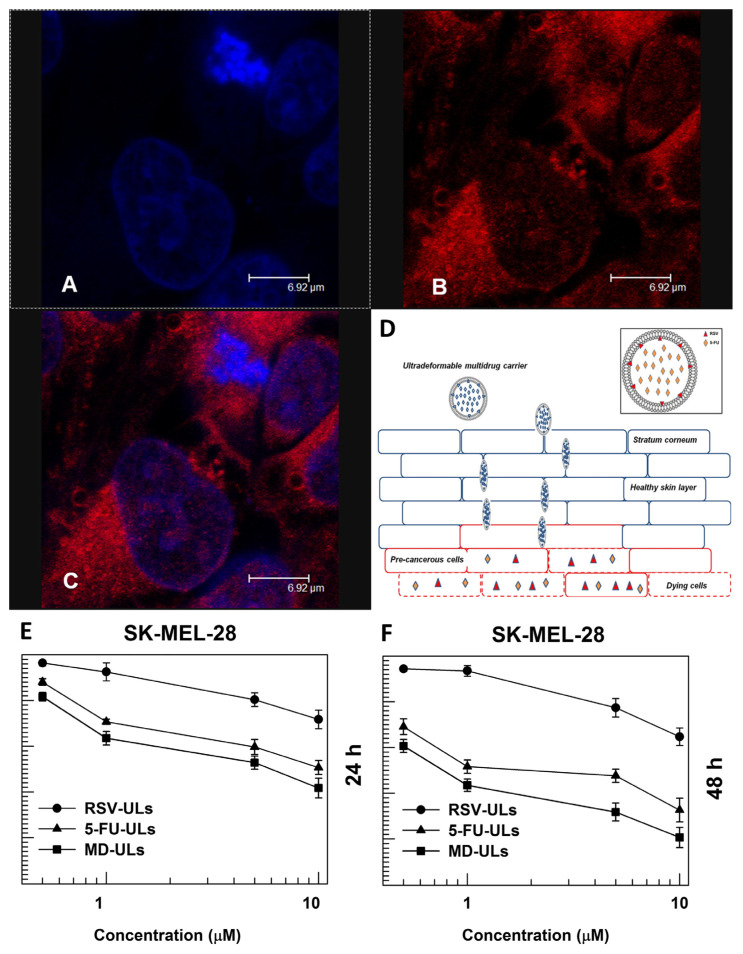
(**A**)—Colo-38 cells’ confocal laser scanning microscopy micrographs, with Hoechst filter, after treatment with rhodamine-labeled ULs; (**B**)—Colo-38 cells’ confocal laser scanning microscopy micrographs, with TRITC filter, after treatment with rhodamine-labeled ULs; (**C**)—Colo-38 cells’ confocal laser scanning microscopy micrographs, after treatment with rhodamine-labeled ULs’ overlay; (**D**)—Representative schematic representation of the developed nanocarriers and their skin permeation capacity; (**E**)—In vitro cytotoxic in SK-MEL-28 cancer cells (melanoma) of the developed vesicles, RSV-ULs, 5-FU-ULs, and MD-ULs after 24-h exposure; (**F**)—In vitro cytotoxic in SK-MEL-28 cancer cells (melanoma) of the developed vesicles, RSV-ULs, 5-FU-ULs, and MD-ULs after 48-h exposure; adapted from Cosco et al. [41].

**Figure 10 pharmaceutics-16-01200-f010:**
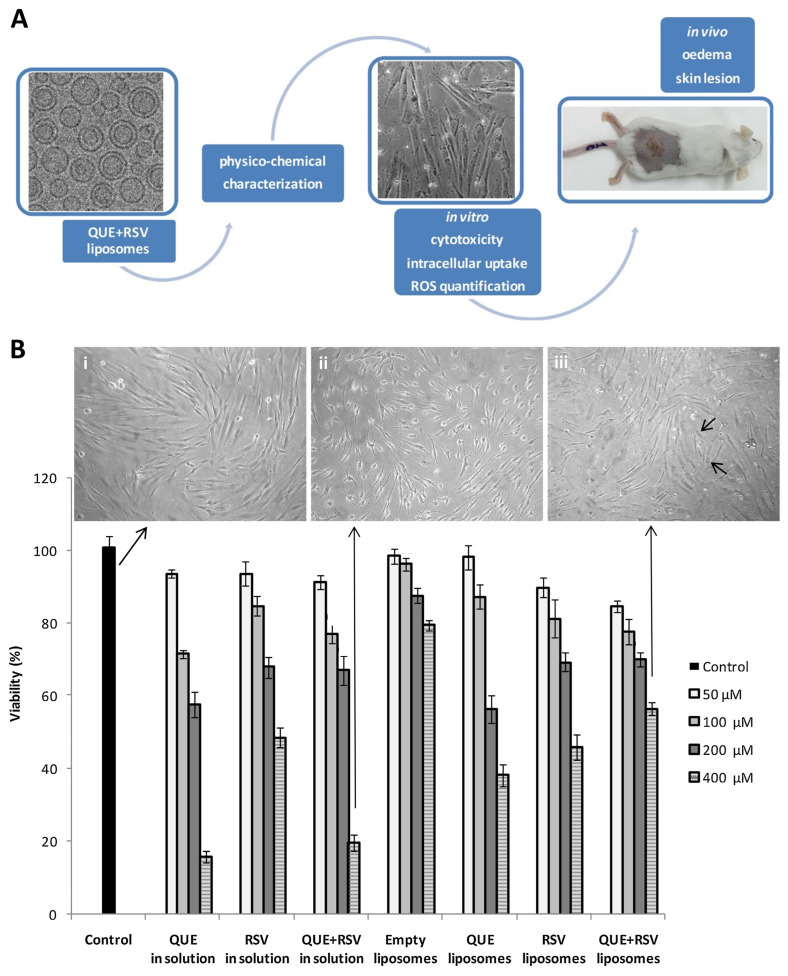
(**A**)—Schematic representation of the produced liposomal vesicles and performed assays; (**B**)—Fibroblast viability results after treatment with the developed liposomal formulations, and controls (24 h exposure), at different quercetin (QUE) and resveratrol (RSV), individually or in combination (perinuclear localization indicated by the small arrows); (**C**)—Evolution of Turbiscan backscattering profiles for empty liposomes (no drugs), quercetin (QUE)-loaded liposomes, resveratrol (RSV)-loaded liposomes, or dual-loaded (QUE + RSV) liposomes, over an 8-day time period, at both 25 and 40 °C; adapted from Caddeo et al. [16].

**Figure 11 pharmaceutics-16-01200-f011:**
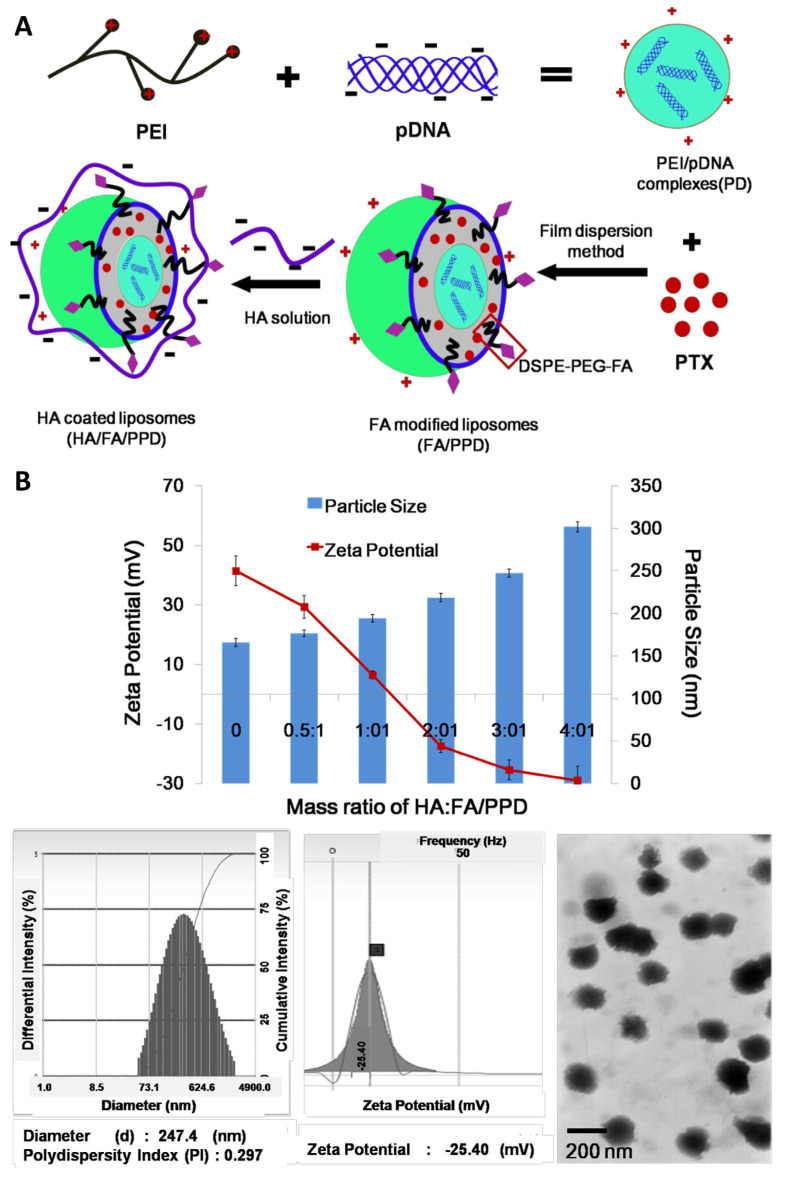
(**A**)—Schematic representation of the developed co-loaded liposomal systems; (**B**)—Particle size and zeta potential of the developed co-loaded liposomal systems; adapted from Liu et al. [1]. (**C**)—In vitro drug release profiles of the developed liposomal systems, with a TEM morphology image of the HA/FA/PPD liposomes after a 12-h incubation with the release media, ** *p* < 0.01, statistically significant difference between FA/PPD and Taxol^®^; ## *p* < 0.01, statistically significant difference between HA/FA/PPD and Taxol^®^; (**D**)—Cytotoxicity levels of different concentrations of the developed liposomal systems on B16 cells, # *p* < 0.05, ** *p* < 0.01; (**E**)—Cellular uptake of the developed liposomal systems, # *p* < 0.05, ## *p* < 0.01; adapted from Liu et al. [1].

**Table 1 pharmaceutics-16-01200-t001:** Most relevant summarized parameters of the analyzed studies regarding dual-loaded liposomal systems for the treatment of skin cancer, including co-loaded drug molecules, particle size, PDI, zeta potential, encapsulation efficiency, main biological findings, and respective references.

Ref.	Co-Loaded Drug Molecules	PS (nm)	PDI	ZP (mV)	EE (%)	Main Biological Findings
[43]	Doxorubicin and celecoxib	142.37 ± 0.78	0.27 ± 0.026	−5.04 ± 0.51	Doxorubicin 98.42 ± 0.0073 Celecoxib 98.37 ± 0.037	Microneedle pre-treatment enhanced drug skin penetration High tumor inhibition Dual drug delivery synergistic effects
[17]	Doxorubicin and ceramide	C6-ceramide 148 ± 10 C8-ceramide 169 ± 18 C8-glucosylceramide 181 ± 10	C6-ceramide 0.131 ± 0.02 C8-ceramide 0.114 ± 0.04 C8-glucosylceramide 0.062 ± 0.01	C6-ceramide 40.8 ± 2.9 C8-ceramide 41.2 ± 3.9 C8-glucosylceramide 35.6 ± 2.5	C6-ceramide 92.86 ± 1.1 C8-ceramide 90.24 ± 1.7 C8-glucosylceramide 92.84 ± 1.4	High cancer cell cytotoxicity Dual drug delivery synergistic effects
[15]	Doxorubicin and hispolon	Doxorubicin 92 ± 1.6 Hispolon 91 ± 2.6	Doxorubicin 0.134 ± 0.12 Hispolon 0.101 ± 0.08	Doxorubicin −44.5 Hispolon −43.2	Doxorubicin 96.54 Hispolon 91.61	Enhanced cancer cell cytotoxicity, with apoptosis Dual drug delivery synergistic effects
[37]	5-fluorouracil and cetuximab	137.0 ± 25	0.26 ± 0.04	−6 ± 1	NM	Increased cellular uptake Increased accumulation in viable epidermis Topical iontophoresis being more effective than subcutaneous treatment (reduced cell proliferation and tumor growth inhibition) Dual drug delivery synergistic effects
[41]	5-fluorouracil and resveratrol	445.6 ± 19.5	0.32 ± 0.09	−25.5 ± 0.5	Resveratrol 97.0 ± 3.2 5-fluorouracil 41.9 ± 1.1	Improved permeation in the skin Dual drug delivery synergistic effects
[16]	Quercetin and resveratrol	79.0 ± 4.1	0.12	−40.0 ± 6.7	Quercetin 71.2 ± 10.9 Resveratrol 72.1 ± 6.6	Anti-ROS activity Tissue renovation and wound healing Dual drug delivery synergistic effects
[1]	Paclitaxel and DNA	247.4 ± 4.2	0.297	−25.40 ± 2.7	NM	Improved transfection efficacy Enhanced cellular uptake Dual drug delivery synergistic effects
[80]	Curcumin and STAT3 siRNA	195.0 ± 9.0	0.240 ± 0.005	58.8 ± 6.0	87.5 ± 4.0	Fast and high cellular uptake Cancer cell growth inhibition Induction of apoptosis in cancer cells Dual drug delivery synergistic effects
[81]	Curcumin and STAT3 siRNA	192.6 ± 9.0	0.326 ± 0.004	56.4 ± 8.0	86.8 ± 6.0	Cancer cell growth inhibition Tumor suppression Dual drug delivery synergistic effects
[12]	Aurora-A inhibitor XY-4 and Bcl-xl siRNA	91.3 ± 4.5	0.183	38.5 ± 0.5	XY-4 84.6	Successful siRNA transfection Enhanced cell uptake and antitumor effect Dual drug delivery synergistic effects
[97]	1-Methyl-tryptophan and cytosine–phosphate–guanosine anionic peptide	453.00 ± 3.80	0.33	NM	99.2	Increased effectiveness in the suppression of tumor growth Dual drug delivery synergistic effects
[19]	Bufalin and anti-CD40 antibody	205.4 ± 68.4	0.062	−15.68	73.59 ± 3.14	Reduced systemic toxicity Enhanced antitumor effect Dual drug delivery synergistic effects

EE—encapsulation efficiency; NM—not mentioned; PDI—polydispersity index; PS—particle size; Ref.—reference; ZP—zeta potential.

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
