# Peer review of "Improving Skin Cancer Treatment by Dual Drug Co-Encapsulation into Liposomal Systems—An Integrated Approach towards Anticancer Synergism and Targeted Delivery"

_pharmaceutics, 2024, doi:10.3390/pharmaceutics16091200_

Round 1

Reviewer 1 Report

Comments and Suggestions for Authors

This manuscript is well-written. However, there are a few areas that could be improved. It would be beneficial to include information on any recent patents that have emerged in the market. Additionally, it would be helpful to add insights on.

Comments on the Quality of English Language

This manuscript is well-written. However, there are a few areas that could be improved. It would be beneficial to include information on any recent patents that have emerged in the market. Additionally, it would be helpful to add insights on.

Author Response

We thank the reviewer for their suggestion, this information has now been added to the introduction section, between lines 328 and 344, supported by the relevant references, which have also been added to the manuscript (changes marked in blue).

Reviewer 2 Report

Comments and Suggestions for Authors

The article is well written and comprehensively summarized. No comments to the authors

Author Response

We thank the reviewer for their substantially positive commentary.

Reviewer 3 Report

Comments and Suggestions for Authors

This review highlights the potential for enhancing skin cancer treatment through dual-loaded liposomal systems. Before publication, the authors should consider the following points:

1. The authors should provide a detailed summary of the mechanisms of action for different liposomal systems, with particular emphasis on the drug release mechanisms. This is especially important for DNA/siRNA, which require intracellular delivery.

2. The authors should also discuss the delivery methods employed by liposomal systems, including whether they rely on passive penetration or are used in conjunction with other technologies such as microneedle pre-treatment or iontophoresis.

Author Response

This review highlights the potential for enhancing skin cancer treatment through dual-loaded liposomal systems. Before publication, the authors should consider the following points:

  1. The authors should provide a detailed summary of the mechanisms of action for different liposomal systems, with particular emphasis on the drug release mechanisms. This is especially important for DNA/siRNA, which require intracellular delivery.

Answer: We thank the reviewer for their comment, this information has now been added to the introduction section, between lines 403 and 416, along with the relevant references, which have also been added to the manuscript (changes marked in blue).

  1. The authors should also discuss the delivery methods employed by liposomal systems, including whether they rely on passive penetration or are used in conjunction with other technologies such as microneedle pre-treatment or iontophoresis.

Answer: We thank the reviewer for their insight, this information has now been added to the introduction section, between lines 417 and 424, along with the applicable references, which have also been added to the manuscript (changes marked in blue).